# Provable Emergence of Deep Neural Collapse and Low-Rank Bias in $L^2$-Regularized Nonlinear Networks

**Emanuele Zangrando**                                          *emanuele.zangrando@gssi.it*
*Gran Sasso Science Institute, L'Aquila, Italy.*

**Piero Deidda**                                                *piero.deidda@sns.it*
*Scuola Normale Superiore, Pisa, Italy.*
*Gran Sasso Science Institute, L'Aquila, Italy.*

**Simone Brugiapaglia**                                  *simone.brugiapaglia@concordia.ca*
*Concordia University, Montréal, Canada.*

**Nicola Guglielmi**                                            *nicola.guglielmi@gssi.it*
*Gran Sasso Science Institute, L'Aquila, Italy.*

**Francesco Tudisco**                                           *f.tudisco@ed.ac.uk*
*Gran Sasso Science Institute, L'Aquila, Italy.*
*University of Edinburgh, Edinburgh, UK.*

**Reviewed on OpenReview:** *https://openreview.net/forum?id=PvmFUzchzY*

## Abstract

We present a unified theoretical framework connecting the first property of Deep Neural Collapse (DNC1) to the emergence of implicit low-rank bias in nonlinear networks trained with $L^2$ weight decay regularization. Our main contributions are threefold. First, we derive a quantitative relation between the Total Cluster Variation (TCV) of intermediate embeddings and the numerical rank of stationary weight matrices. In particular, we establish that, at any critical point, the distance from a weight matrix to the set of rank-$K$ matrices is bounded by a constant times the TCV of earlier-layer features, scaled inversely with the weight-decay parameter. Second, for invertible nonlinearities, we prove global optimality of DNC1 in a constrained representation-cost setting for both feedforward and residual architectures, showing that zero TCV across intermediate layers minimizes the representation cost under natural architectural constraints. Third, we establish a benign landscape property: for almost every interpolating initialization there exists a continuous, loss-decreasing path from the initialization to a globally optimal, DNC1-satisfying configuration. Our theoretical claims are validated empirically; numerical experiments confirm the predicted relations among TCV, singular-value structure, and weight decay. These results indicate that neural collapse and low-rank bias are intimately linked phenomena arising from the optimization geometry induced by weight decay.

## 1 Introduction

Recent years have seen growing theoretical and experimental evidence of low-rank properties of deep neural networks. For example, it has been noted that a large portion of singular values from linear layers in different models can be removed almost without affecting performance, both for feedforward and residual networks (Schotthöfer et al., 2022; Idelbayev & Carreira-Perpiñán, 2020). Analogous results have been shown on feedforward and convolutional networks for image classification (Li et al., 2019; Wang et al., 2021; Khodak et al., 2021; Schotthöfer et al., 2022) as well as on Recurrent Neural Networks (RNNs) trained to learn neural dynamics in neuroscience tasks (Schuessler et al., 2020; Pellegrino et al., 2023).

Alongside the wealth of experimental evidence of low-rank properties of neural networks, several recent works have investigated the low-rank bias phenomenon from the theoretical point of view, providing evidence of decaying singular values in deep linear architectures (Arora et al., 2019; Feng et al., 2022; Huh et al., 2022) and of low-rank properties of networks' Hessian maps (Singh et al., 2021), analyzing low-rank geometric properties of training gradient flow dynamics for deep linear networks (Bah et al., 2022), showing that low-rank bias is related to weight decay (Galanti et al., 2023; Súkeník et al., 2024a; Kuzborskij & Yadkori, 2025), and to training invariances (Le & Jegelka, 2022; Ziyin et al., 2025). In particular, the authors relate the rank of the weight matrices during SGD to the batch size adopted during the training of the network. However, based only on the batch size, the results from (Galanti et al., 2023) become uninformative when the batch size is comparable to the number of parameters of the neural network.

A related phenomenon commonly observed during the training of classifiers is "neural collapse" (Papyan et al., 2020; Zhu et al., 2021) and more generally "deep (or intermediate) neural collapse" Parker et al. (2023); Rangamani et al. (2023b); Súkeník et al. (2024b;a). This phenomenon is rigorously described in the literature using four properties named **NC1–4**, the first of which (**NC1**) essentially states that the within-class variability of intermediate representations goes to zero even for layers in the middle of a deep neural network (see Definition 3.3). The optimality of neural collapse at the last layer has been extensively investigated and proved in different settings, otherwise only a few extensions have been proposed for its deep version. In this work, we theoretically investigate the phenomenon of deep neural collapse together with its connection with the low-rank implicit bias on a large class of nonlinear networks in presence of $L^2$ regularization.

## 1.1 Related work

The emergence of the neural collapse phenomenon at the last layer was first empirically observed in (Papyan et al., 2020), and proven to be optimal in different settings, such as Unconstrained Feature Model (UFM) (Fang et al., 2021; Mixon et al., 2020; Zhu et al., 2021), with constrained UFM (Dang et al., 2024; Yaras et al., 2023), with cross-entropy loss (E & Wojtowytsch, 2021; Kunin et al., 2023; Hong & Ling, 2024), Neural Tangent Kernel (NTK) block structure aligned with class labels (Seleznova et al., 2023), linear tail and wide architecture (Jacot et al., 2025), effects of the loss function (Zhou et al., 2022b), and in the context of Graph Neural Networks (GNNs) (Kothapalli et al., 2023). Moreover, some works analyzed the dynamics leading to neural collapse, from the observation of a central path dynamics (Han et al., 2022), to the use of large network dynamics (Jacot et al., 2025). Finally, some authors characterize stability of the stationary points of UFM in the regularized setting (Ji et al., 2022; Zhu et al., 2021; Zhou et al., 2022a), and in the Riemannian setting, with feature and weights constrained to oblique manifolds (Yaras et al., 2023).

The analysis of the neural collapse phenomenon beyond the last layer was first empirically studied by (Rangamani et al., 2023b), and addressed under the name of "deep" (or "intermediate") neural collapse phenomenon (**DNC**), where notably the NC1 property is observed to hold even at intermediate layers. Proving DNC1 optimality has proven more challenging than NC because it requires switching from the linear setting of the last layer to the nonlinear one of the intermediate layers. A theoretical analysis of this phenomenon has been conducted in (Súkeník et al., 2024b;a) that establish DNC1 optimality in the Deep-UFM model but restrict to binary classification or require specific architectural assumptions (L-DUFM, pyramidal widths). In particular, the key limitation of prior work about DNC1 is that it studies only simplified linear/UFM models, not able to take into account the full extent of architectural constraints.

## 1.2 Main Contributions

In this work we show, for the first time, a rigorous connection between the phenomenon of deep neural collapse and low-rank implicit bias for general nonlinear networks. The main contribution is theoretical and, in particular, our main results, *which hold in the presence of weight decay*, can be summarized as follows:

- in Section 3 we prove that within-class variability of hidden layers during training (including the initial data set) has a quantifiable effect on the numerical rank of the network parameters as the latter decays with the total cluster variation of the hidden representations (Theorem 3.5);

- in Section 4 we show that zero total cluster variation for all intermediate layers is optimal in a fairly general setting, *both for feedforward and residual networks* assuming invertible nonlinearities (Theorem 4.2);

- Theorem 4.4 establishes that for (almost) every interpolating initial condition, there exists a loss-decreasing trajectory connecting the initial condition to a collapsed configuration, thereby offering an explanation to the consistent empirical observation of deep neural collapse (Rangamani et al., 2023b).

Overall, the combination of the results above proves that in the regime of small weight decay regularization, zero within-class variability for all intermediate representations is globally optimal under some natural constraints, therefore showing that global minima in parameter space are low-rank matrices. Moreover, convergence to these highly structured solutions may be regarded as universal in the sense that, *there is essentially no loss barrier between interpolating features to deep-neural collapsed ones*, which constitutes a step forward towards explaining the universality in the emergence of (deep) neural collapse.

Our paper improves upon the results presented in the related literature in different aspects. First, we show that the rank of stationary points in parameter space is controlled by DNC1, a metric defining the first property of neural collapse. In particular, our result also holds for stationary points that do not satisfy DNC2/DNC3. Notably, the results in (Súkeník et al., 2024a) suggest that there exist settings where our results become crucial, indeed the authors show that, in such settings, while DNC1 keeps being optimal, the other properties may fail.

Second, the setting in which we prove our results is not a simplified model, but an exact representation of a feedforward/ResNet architecture with two additional constraints. The analysis is based on the extrinsic view of a neural network architecture as a constraint in the space of representable intermediate embeddings (Jacot et al., 2022; Han et al., 2022). Finally, in this generalized setting, we prove a *benign* property of the loss landscape, which could explain the frequent emergence of neural collapse in intermediate representations as observed in (Rangamani et al., 2023b).

The remainder of the paper is organized as follows. In Section 2 we introduce notation and set up the optimization framework. In Section 3 we prove the rank–TCV bound and discusses its implications. Section 4 contains the optimality and landscape results, and Section 5 validates the theory empirically.

## 2 Setting and Preliminaries

Consider a dataset $\mathcal{X} = \{x_j\}_{j=1}^N \subseteq \mathbb{R}^d$ with $c$ corresponding labels $\mathcal{Y} = \{y_j\}_{j=1}^N \subseteq \mathbb{R}^c$.

We assume to train a fully connected $L$-layer feedforward neural network $f(\Theta, x)$ of the form

$$z^0 = x, \quad f(\Theta, x) = z^L, \quad z^l = \sigma_l(W_l z^{l-1}), \; l = 1, \dots, L \tag{1}$$

to learn from the given data and labels. In equation equation 1, $\sigma_l$ are scalar nonlinear activation functions acting entrywise and $\Theta$ denotes all trainable parameters $\Theta = (W_1, \dots, W_L)$. We write

$$\begin{aligned}
\Theta_{l_1}^{l_2} &:= (W_{l_1+1}, \dots, W_{l_2}), \quad l_1 < l_2, \\
f_{l_1}^{l_2}(\Theta_{l_1}^{l_2}, z^{l_1}) &:= \sigma(W_{l_2} \cdots \sigma(W_{l_1+1} z^{l_1}))
\end{aligned} \tag{2}$$

to denote the training parameters and concatenation of layers from $l_1 + 1$ to $l_2$, respectively. We adopt the convention that, when we do not specify the lower index then it starts from zero, and when we omit the top one it ends at the overall depth $L$, namely, $\Theta^{l_2} := \Theta_0^{l_2}$, $f^{l_2} := f_0^{l_2}$ and $\Theta_{l_1} := \Theta_{l_1}^L$, $f_{l_1} := f_{l_1}^L$. Notice that, with this notation, $\Theta = \Theta_0 = \Theta^L = \Theta_0^L$, $f = f_0 = f^L = f_0^L$ and $\Theta^0$ is empty. Finally, for the sake of brevity, we write $\mathcal{Z}^l := \{z_i^l\}_{x_i \in \mathcal{X}}$ to denote the set of outputs of the $l$-th layer, where $z_i^l = f^l(\Theta^l, x_i)$ is defined as in equation 1 with input data $x_i$, and $\mathcal{Z}^0 = \mathcal{X}$.

Moreover, to simplify notation, we will denote the feature matrices of input, output, and intermediate layers as

$$X = [x_1, \dots, x_N] \in \mathbb{R}^{d \times N}, \quad Y = [y_1, \dots, y_N] \in \mathbb{R}^{c \times N}, \quad Z^l = [z_1^l, \dots, z_N^l] \in \mathbb{R}^{n_l \times N},$$

where $1 \le l \le L$. The parameters $\Theta$ are assumed to be trained by minimizing a regularized loss function $\mathcal{L}_\lambda(\Theta)$ composed by the sum of losses at individual data points, i.e.,

$$\mathcal{L}_\lambda(\Theta) := \frac{1}{N} \sum_{i=1}^N \ell\left(f(\Theta, x_i), y_i\right) + \lambda \|\Theta\|^2, \quad \text{where} \quad \|\Theta\|^2 = \sum_{i=1}^L \|W_i\|^2. \tag{3}$$

Here $\|\cdot\|$ denotes the entrywise $L^2$ norm (when the input is a matrix, it coincides with the Frobenius norm) and $\lambda \ge 0$ is a *weight decay* or *regularization* parameter.

In this setting, in Section 3 we will quantify a precise relationship between the numerical rank of the stationary points of $\mathcal{L}_\lambda$ and the total cluster variation of intermediate feature matrices.

## 3 A Link Between Low-Rank and Deep-Neural Collapse

Note that any critical point of $\mathcal{L}_\lambda(\Theta)$ satisfies the following equation:

$$\lambda W_l^* = -\nabla_{W_l} \mathcal{L}_0(\Theta^*) \qquad \forall l = 1, \dots, L. \tag{4}$$

The intuition is that Equation equation 4 resembles an "eigenvalue equation" for the nonlinear operator $\nabla \mathcal{L}_0$. In other words, for a fixed weight decay parameter $\lambda$, training the neural network corresponds to finding a nonlinear eigenvector $\Theta^*$ having $-\lambda$ as eigenvalue. In addition, equation 4 immediately implies that whenever the operators $\nabla_{W_l} \mathcal{L}_0$ have low rank (i.e., they take values in some low-rank space), then also any weight matrix $W^*$ composing the eigenvector $\Theta^*$ has necessarily low rank. In the following, we investigate this relationship more in depth, providing sufficient conditions to have approximately low-rank gradients.

The first key point in this study is provided by the following proposition. Indeed, for any differentiable loss function $\ell$, the gradient with respect to any layer, at a fixed data point, has rank at most 1 (see also Lemma 3.1 in (Galanti et al., 2023)). Here we consider the gradient at a batch of data points. The proof of this result can be found in Appendix A.1.

**Proposition 3.1** (Small batches yield low-rank gradients)**.** *Let $f(\Theta, x)$ be a feedforward neural network as in equation 1 and $\ell : \mathbb{R}^c \times \mathbb{R}^c \to \mathbb{R}$ be a differentiable loss function as in equation 3. Then, for any $\{(x_i, y_i)\}_{i=1}^K \subseteq \mathbb{R}^d \times \mathbb{R}^c$, we have $\mathrm{rank}\left(\nabla_{W_l} \sum_{i=1}^K \ell\left(f(\Theta, x_i), y_i\right)\right) \le K$.*

Note that, given $z^j = f^j(\Theta^j, x)$, as long as $j < l$, it holds

$$\nabla_{W_l} \ell\left(f(\Theta, x), y\right) = \nabla_{W_l} \ell\left(f_j(\Theta_j, z^j), y\right). \tag{5}$$

Hence, fixed a batch of data, Proposition 3.1 allows us to control the rank of the gradient, with respect to a subsequent layer, of any intermediate loss function $\ell$.

In addition, Proposition 3.1 immediately yields an example where gradients have necessarily low rank. Consider the case of degenerate data, i.e., assume that there exist $K$ pairs $\{(\bar{x}_k, \bar{y}_k)\}_{k=1}^K$ such that any data point $(x_i, y_i)$ is equal to some $(\bar{x}_k, \bar{y}_k)$. Then, by Proposition 3.1, $\nabla_{W_l} \sum_i \ell(\Theta, x_i, y_i)$ has rank at most $K$. But the right-hand side of equation 4 has rank at most $K$, hence any critical point of the loss function $\mathcal{L}_\lambda$ necessarily corresponds to a neural network having all layers of rank at most $K$. The same argument can be extended to the situation of total intermediate neural collapse (Rangamani et al., 2023a), where the intermediate hidden set $\mathcal{Z}^l$ consists of at most $K$ distinct points. Indeed, assume that for a stationary point $\Theta^*$ there exists a layer $l$ and points $\{(\bar{z}_k^l, \bar{y}_k)\}_{k=1}^K$ such that any pair $(z_i^l, y_i) = (\bar{z}_k^l, \bar{y}_k)$ for some $k$. Then, by Proposition 3.1 and the subsequent remark equation 5, any subsequent layer, $W_j^*$ with $j > l$, has rank at most $K$.

In the following, we prove that under weaker hypotheses, like the collapse of hidden representations, any critical point of the loss function produces layers close to low-rank. In particular, the approximate rank of the layers at a stationary point is given by the number of clusters (or collapsing sets) and the approximation error depends both on the regularity of the loss function in the clusters and the within-cluster variability, which we measure as the total cluster variation of each hidden set $\mathcal{Z}^l$.

### 3.1 Clustered Data and Total Cluster Variation

Assume $\mathcal{C} = \{\mathcal{C}_1, \dots, \mathcal{C}_K\}$ to be a $K$-partition of the input data $\mathcal{X}$ and corresponding labels $\mathcal{Y}$, i.e., for any $(x_i, y_i) \in (\mathcal{X}, \mathcal{Y})$ there exists a unique $\mathcal{C}_k \in \mathcal{C}$ such that $(x_i, y_i) \in \mathcal{C}_k$. We denote by $N_k = |\mathcal{C}_k|$ the cardinality of the $k$-th family and adopt the notation $(\mathcal{X}, \mathcal{Y}) = \bigcup_{k=1}^{K} \bigcup_{i \in \mathcal{C}_k} \{(x_i, y_i)\} = \bigcup_{k=1}^{K} \bigcup_{i=1}^{N_k} \{(x_i^k, y_i^k)\}$, where with a small abuse of notation we use $\mathcal{C}_k$ to denote also the set of indices whose corresponding data points are contained in $\mathcal{C}_k$. Now let $\bar{x}_k = \bar{z}_k^0$, $\bar{y}_k$ and $\bar{z}_k^l$ be the centroids of the sets $\{x_i\}_{i \in \mathcal{C}_k}$, $\{y_i\}_{i \in \mathcal{C}_k}$ and $\{z_i^l\}_{i \in \mathcal{C}_k}$, respectively, i.e.,

$$\bar{z}_k^l = \frac{1}{N_K} \sum_{i \in \mathcal{C}_k} z_i^l, \qquad \bar{y}_k = \frac{1}{N_K} \sum_{i \in \mathcal{C}_k} y_i.$$

To quantify how clustered the data is with respect to the partition $\mathcal{C}$, we consider the *Within-Cluster Sum of Squares (WCSS)*, defined as

$$\text{WCSS}_l(\mathcal{C}) = \frac{1}{N} \sum_{k=1}^{K} \sum_{i \in \mathcal{C}_k} \|(z_i^l, y_i) - (\bar{z}_k^l, \bar{y}_k)\|^2. \tag{6}$$

This quantity measures the within-cluster variance (see, e.g., (Hastie et al., 2009)) of the data at layer $l$ with respect to a prescribed partition and the corresponding centroids.

Note that having a clustered dataset at the $l$-th layer means that there exists a suitable partition $\mathcal{C}^*$ such that $\text{WCSS}_l(\mathcal{C}^*)$ is sufficiently small. To measure the within-cluster variability of the data at the $l$-th layer when it is partitioned into at most $K$ clusters, we introduce the *Total Cluster Variation*.

**Definition 3.2** (Total Cluster Variation). The $K$-th *Total Cluster Variation (TCV)* at layer $l$, is defined as

$$\text{TCV}_{K,l} := \min_{\mathcal{C} \in \cup_{r=1}^{K} \mathcal{P}_r} \text{WCSS}_l(\mathcal{C}), \tag{7}$$

where $\mathcal{P}_r$ is the set of all partitions of $\{(z_i^l, y_i)\}_{i=1}^{N}$ into $r$ sets.

Note that, for $K$ sufficiently smaller than $N$, a small value of $\text{TCV}_{K,l}$ indicates an intermediate collapse of the network's features at layer $l$. Moreover, being $\text{TCV}_{K,l}$ defined through a variational problem to get the tightest bound possible, it is trivially upper bounded by the total cluster variation on the classes. These considerations are directly connected with the notion of deep (or intermediate) neural collapse (Súkeník et al., 2024b; Papyan, 2020; Rangamani et al., 2023b), whose first characterizing property is recalled below.

**Definition 3.3** (Deep Neural Collapse 1). We say that a neural network $f$ trained on a dataset $(\mathcal{X}, \mathcal{Y})$ satisfies DNC1 at layer $l$ if $\text{TCV}_{K,l} = 0$ for $K = c$ and the minimizer in equation 7 is given by the partition corresponding to the classes.

### 3.2 Neural Collapse Yields Low-Rank Bias

In this section, we show that the occurrence of well-clustered hidden representations, i.e., of intermediate neural collapse, necessarily yields low-rank properties for the underlying neural network in the presence of weight decay. Given a partition $\mathcal{C} = \{\mathcal{C}_k\}_{k=1}^{K}$ as in Section 3.1, we can introduce the centroid-based loss at the $l$-th layer:

$$\mathcal{L}_\lambda^{l,\mathcal{C}}(\Theta) = \sum_{k=1}^{K} \pi_k \, \ell\left(f_l(\Theta_l, \bar{z}_k^l), \bar{y}_k\right) + \lambda \|\Theta\|^2, \quad \pi_k = \frac{N_k}{N}.$$

Note that $\mathcal{L}_\lambda^{0,\mathcal{C}}$ is equivalent to the standard loss $\mathcal{L}_\lambda$ in the situation of degenerate data, i.e., when the data within any family of $\mathcal{C}$ coincides with its centroid. Moreover, from the previous discussion and Proposition 3.1, we know that the gradient with respect to any of the last $L - l$ layers of the $l$-th centroid-based loss has rank at most $K$. In the following lemma (proved in Appendix A.2), an easy computation allows us to compare the gradients of the original loss and the centroid-based loss:

**Lemma 3.4** (Centroid-based gradient approximation of full gradient). *Fix a partition $\mathcal{C} = \{\mathcal{C}_k\}_{k=1}^K$ and a layer $l$ with weight matrix $W_l$. Let the function $\ell(f_j(\Theta_j, z), y)$ be $C^1$ in $\Theta_j$ and $\nabla_{W_l}\ell(f_j(\Theta_j, z), y)$ be $C^2$ in $(z, y)$ for any $j < l$. Then, $\forall j < l$*

$$\nabla_{W_l}\mathcal{L}_0(\Theta) = \nabla_{W_l}\mathcal{L}_0^{j,\mathcal{C}}(\Theta) + R(\Theta), \quad \|R(\Theta)\| \leq M_{l,j}(\Theta, \mathcal{C})\,\mathrm{WCSS}_j(\mathcal{C})$$

*where $M_{l,j}(\Theta, \mathcal{C}) := \sup\limits_{\substack{k=1,\ldots,K \\ (z,y)\in\mathcal{H}_k^j}} \|\nabla_{(z,y)}^2 \nabla_{W_l}\ell(f_j(\Theta_j, z), y)\|$ and $\mathcal{H}_k^j := \mathrm{ConvHull}\{(z_i^j, y_i)\}_{i\in\mathcal{C}_k}$.*

*Proof (sketch).* Here we provide a sketch of the proof and we refer to Appendix A.2 for a detailed argument. Fixed a single data point $(z_i^j, y_i)$ with $i \in \mathcal{C}_k$, we consider the Taylor expansion in $(\bar{z}_k^j, \bar{y}_k)$ of $\nabla_{W_l}\ell(f_j(\Theta_j, z_i^j), y_i)$ with Lagrange remainder. Finally, we note that the first-order contributions sum to zero because of the definitions of the centroids, yielding the desired expressions. $\qquad\square$

As a consequence of the above lemma, for any stationary point, we can control the distance of the $l$-th layer from the set of matrices having rank less than or equal to $K$ in terms of the $K$-th total cluster variation of the first $l-1$ intermediate outputs. Before stating our main result, we define the set of $n_1 \times n_2$ matrices having rank equal to $K$, $\mathcal{M}_K^{n_1, n_2} := \{A \in \mathbb{R}^{n_1\times n_2} \mid \mathrm{rank}(A) = K\}$. We will omit the dimension-related superscripts when clear from the context.

**Theorem 3.5** (Small within-class variability yields low-rank bias). *Assume $\Theta^*$ to be a stationary point of $\mathcal{L}_\lambda(\Theta)$. Then, for any $l = 1, \ldots, L$ and $K \leq N$,*

$$\min_{Z\in\cup_{r=1}^K\mathcal{M}_r}\|W_l^* - Z\| \leq \min_{j=1,\ldots,l-1}\frac{M_{l,j}(\Theta^*)}{\lambda}\mathrm{TCV}_{K,j},$$

*where $M_{l,j}(\Theta^*) := \sup\limits_{(\alpha,\beta)\in\mathcal{H}^j}\|\nabla_{(z_j,y)}^2\nabla_{W_l}\ell(f_j(\Theta_j^*, \alpha), \beta)\|$ and $\mathcal{H}^j := \mathrm{ConvHull}\{(z_i^j, y_i)\}_{i=1}^N$.*

*Proof (sketch).* Observe that from equation 4, at the stationary point we have

$$W_l^* = -\frac{1}{\lambda}\nabla_{W_l}\mathcal{L}_0(\Theta^*). \tag{8}$$

Now, Lemma 3.4 allows us to control the distance of the gradient in the right-hand side of equation 14 from $\mathcal{M}_K^{n_{l-1}, n_l}$. In particular, given $\mathcal{C} \in \cup_{r=1}^K\mathcal{P}_r$, for any $j = 1, \ldots, l-1$ we see that

$$\min_{Z\in\cup_{r=1}^K\mathcal{M}_r}\|W_l^* - Z\| \leq \frac{\|\nabla_{W_l}\mathcal{L}_0(\Theta^*) - \nabla_{W_l}\mathcal{L}_0^{j,\mathcal{C}}(\Theta^*)\|}{\lambda} \leq \frac{M_{l,j}(\Theta^*, \mathcal{C})\,\mathrm{WCSS}_j(\mathcal{C})}{\lambda},$$

where we have used that $\nabla_{W_l}\mathcal{L}_0^{j,\mathcal{C}}(\Theta^*)/\lambda$ has rank smaller than or equal to the number of families of $\mathcal{C}$. Then, first minimizing over $j = 1, \ldots, l-1$, second bounding from above $M_{l,j}(\Theta^*, \mathcal{C}) \leq M_{l,j}(\Theta^*)$ for any $j$, and third minimizing over all the possible partitions $\mathcal{C} \in \cup_{r=1}^K\mathcal{P}_r$ yield the thesis. $\qquad\square$

The result in Theorem 3.5, as a low-rank statement, is informative in the regime $K \ll n$, where $n = \min\{n_{l+1}, n_l\}$ is the minimal size of the matrix $W_l$. In particular, one could formalize the notion of low-rank as follows.

**Definition 3.6** (Asymptotically low-rank family of matrices). Consider a family of matrices $X_n \in \mathbb{R}^{n\times n}$. We say that the family $(X_n)_{n\in\mathbb{N}}$ is asymptotically low-rank if $\mathrm{rank}(X_n) = o(n)$ for $n \to +\infty$, i.e., if $\frac{\mathrm{rank}(X_n)}{n} \underset{n\to+\infty}{\to} 0$. We say that the family is asymptotically $\varepsilon$-approximately of rank $r_n$ if $\lim_{n\to+\infty}\mathrm{dist}(X_n, \mathcal{M}_{\leq r_n}) \leq \varepsilon$. If additionally $r_n = o(n)$, we say that the family is asymptotically $\varepsilon$-approximately low-rank.

*Remark* 3.7 (Low-rank in different regimes). Despite the original statement in Theorem 3.5 is stronger (being non-asymptotic), according to Definition 3.6, Theorem 3.5 can be restated as a statement on the family $(W_l^{n,*})_n$ being asympotically $\varepsilon$-approximately low-rank, where $\varepsilon$ is an upper bound on $\min_{j=1,\dots,l-1} \frac{M_{l,j}(\Theta^{n,*})}{\lambda} \text{TCV}_{K,j}^n$, with asymptotical rank $r_n \equiv K$. Combined with the results in Section 4, where we show that for $n \geq N$ we have $\text{TCV}_{K,j} = 0$ on global minimizers, this implies that the family $(W_l^{n,*})_n$ of global minimizers is asymptotically low-rank, in particular with $\text{rank}(W_l^{n,*}) \equiv K = O(1)$.

In particular, Theorem 3.5 shows that intermediate neural collapse (Definition 3.3), combined with weight decay, leads to approximately low-rank network layers, with numerical rank controlled by the TCV of intermediate representations. Its proof can be found in Appendix A.3.

By weight decay, global minimizers lie in a compact set; under smoothness assumptions, this implies uniform boundedness of $M_{l,j}$. We'd like to clarify this in the following remark.

*Remark* 3.8 (Implication of Theorem 3.5 on global minimizers of $\mathcal{L}_\lambda$). The quantity $M_{l,j}(\Theta)$ is uniformly bounded on the set of global minimizer $\Theta^* = \Theta_\lambda^*$ of $\mathcal{L}_\lambda$. In fact, consider any global minimizer $\Theta_\lambda^*$ of $\mathcal{L}_\lambda$, and $\Theta_0^*$ global minimizer of $\mathcal{L}_0$. Then, we have

$$\inf_\Theta \mathcal{L}(\Theta) + \lambda \|\Theta_\lambda^*\|^2 \leq \mathcal{L}_\lambda(\Theta_\lambda^*) \leq \mathcal{L}_\lambda(\Theta_0^*) = \inf_\Theta \mathcal{L}(\Theta) + \lambda \|\Theta_0^*\|^2,$$

which implies $\|\Theta_\lambda^*\| \leq \|\Theta_0^*\|$. In particular, defined $\Omega = \{\Theta \mid \|\Theta\| \leq \|\Theta_0^*\|\}$ (which is compact and contains the set of all global minimizers of $\mathcal{L}_\lambda$), under the regularity hypothesis that the function $(z,y) \mapsto \nabla_{W_l} \ell(f_j(\Theta_j, z), y)$ is of class $C^2$ we obtain that $H^j = H^j(\Theta)$ is bounded, being contained in the ball $\mathcal{K}$ of center zero and radius $\max_{\Theta \in \Omega} \max_{i=1,\dots,N} \|(z_i^j, y_i)\|$ (where the radius is finite thanks to compactness of $\Omega$ and continuity of $f_j$). Therefore, we have

$$M_{l,j}(\Theta^*) \leq \sup_{\Theta \in \Omega} M_{l,j}(\Theta) = \sup_{\Theta \in \Omega} \sup_{(\alpha,\beta) \in H^j(\Theta)} \|\nabla^2_{(z_j,y)} \nabla_{W_l} \ell(f_j(\Theta, \alpha), \beta)\| \leq$$

$$\leq \sup_{\Theta \in \Omega} \sup_{(\alpha,\beta) \in \mathcal{K}} \|\nabla^2_{(z_j,y)} \nabla_{W_l} \ell(f_j(\Theta, \alpha), \beta)\| < +\infty,$$

where the last term is finite thanks to compactness of $\Omega \times \mathcal{K}$ and continuity of $(\Theta, \alpha, \beta) \mapsto \nabla^2_{(z_j,y)} \nabla_{W_l} \ell(f_j(\Theta, \alpha), \beta)$. We further observe that all the conditions above are satisfied if all the activations used are smooth.

Notice that the vanishing property of TCV in intermediate layers (DNC1) has already been studied for simplified models in, e.g., (Rangamani et al., 2023a; Súkeník et al., 2024b;a; Jacot et al., 2025). We therefore notice here the special role of DNC1 with respect to the other DNC properties commonly studied, such as the ones presented in (Súkeník et al., 2024b): the first property alone implies that the stationary points in parameter space must have low-rank, as predicted by Theorem 3.5.

In the next section, we will prove that, in a fairly general setting, DNC1 is optimal for all intermediate layers: this means that under a set of natural constraints, a zero total cluster variation for all intermediate layers is globally optimal in the small $\lambda$ regime. This shows, in combination with Theorem 3.5, that all global minima in this regime are low rank. Moreover, we will show that for almost every initial condition, there is essentially no loss barrier to configurations with $\text{TCV} = 0$ for all $l = 1, \dots, L$, shedding new light on the emergence deep neural collapse and, in turn, on the emergence of low-rank bias.

## 4 Vanishing TCV: Global Optimality of Deep-Neural Collapse

We start by defining the *representation cost* functional for a biasless network with $L^2$ regularization in all layers but the first one:

$$\mathcal{L}(\Theta) := \begin{cases} +\infty, & \text{if } f(\Theta; X) \neq Y, \\ \frac{1}{2} \sum_{l=2}^{L-1} \|W_{l+1}\|^2, & \text{otherwise.} \end{cases} \tag{9}$$

This functional has been studied in a variety of different settings, both linear (Dai et al., 2021) and nonlinear (Jacot, 2023a;b). While in the deep linear case the explicit form of the minima of $\mathcal{L}$ are known to be Shatten

quasi-norms (Dai et al., 2021; Ongie & Willett, 2022), in the general case a full characterization is still missing. In (Jacot, 2023b;a), the authors were able to characterize, in the large depth regime, how the minimal value essentially depends on the data $y = f^*(x)$ one is interpolating, essentially showing that the first order term scales with $L$ and it is a function bounded between the Jacobian and the bottleneck rank of the function $f^*$.

The general intuition is that $\mathscr{L}(\Theta)$ represents the loss of interpolating networks in the small $\lambda$ regime, in fact from (Scagliotti, 2022), we have that $\lambda^{-1}\mathcal{L}_\lambda \underset{\lambda \to 0}{\to} \mathscr{L}(\Theta)$ in the $\Gamma$-convergence sense (Definition A.1), meaning that minimizers of $\mathcal{L}_\lambda$ converge to minimizers of $\mathscr{L}$ as $\lambda$ goes to zero, see Proposition A.2. Now, using the observation done in (Jacot et al., 2022), we can reparameterize minimal norm weight matrices $\{W_l\}$ using *representable* feature matrices in the following way:

$$Z^{l+1} = \sigma_{l+1}(W_{l+1}Z^l) \iff W_{l+1} = \sigma_{l+1}^{-1}(Z^{l+1})Z^{l,+}, \tag{10}$$

where $Z^{l,+}$ denotes the Moore-Penrose pseudoinverse of the feature matrix $Z^l$. We note that the features in equation 10 must be representable: in fact, if we want to reformulate the representation cost function in equation 9 based on the equivalence discussed in equation 10 we have to additionally take into account for a constraint relating the different intermediate representations, literally $Z^l$ and $Z^{l+1}$ have to satisfy the following equation

$$\sigma_{l+1}^{-1}(Z^{l+1})Z^{l,+}Z^l = \sigma_{l+1}^{-1}(Z^{l+1}) \qquad \forall l = 0, \dots, L-1. \tag{11}$$

Note that the constraint in equation 11 is effectively a representability condition on the features, equivalent to $W_{l+1}Z^l = \sigma_{l+1}^{-1}(Z^{l+1})$. Given this new representation of the weight matrices, we can reformulate the representation cost as a function of the feature matrices $Z^l$, namely:

$$\mathscr{L}(Z) := \begin{cases} +\infty, & \text{if } Z^L \neq Y, \\ \frac{1}{2}\sum_{l=2}^{L-1} \|\sigma_{l+1}^{-1}(Z^{l+1})Z^{l,+}\|^2, & \text{otherwise.} \end{cases} \tag{12}$$

As proven in (Jacot et al., 2022), it is not restrictive to study this reparametrized representation cost in the sense that the local minimizers of equation 9, subject to the representability constraints 11, are in bijection with those of equation 12, therefore making them equivalent. The key observation is that the objective in equation 12 is local, i.e., it is a sum of local objectives $\mathscr{L}_l = \mathscr{L}_l(Z^{l+1}, Z^l)$ depending just on neighbour representations. Therefore, since $\|W_1\|^2$ is not present in the regularization, $\mathscr{L}$ depends on $Z^1$ just through $\mathscr{L}_1(Z^2, Z^1) = \frac{1}{2}\|\sigma_2^{-1}(Z^2)Z^{1,+}\|^2$, which can be minimized explicitly with respect to $Z^1$, as a function of the next representation $Z^2$ (and the same holds for the second term). For this reason, the lack of the first two weight matrices in the regularization term is the key to obtaining the results in the following section, making it possible to recursively study the structure of minima in closed form, layer by layer. In particular, the idea is that the minimizers of this modified functional can be described exactly in closed form (Theorem 4.2), and their stability can be studied perturbatively (Theorem 4.6). To study it, we consider the additional hypothesis that $Y$ is in one-hot encoding form, i.e. $y_j = e_{i_j}$ for any $j$, where $\{e_i\}$ is the canonical basis of $\mathbb{R}^c$. We also point out that, without considering additional constraints, the minima of $\mathscr{L}$ may not exist. This can be easily observed in the case in which an activation function $\sigma_2$ is homogeneous. In this case, assume by contradiction that $Z$ is a minimizer and consider the parametric linear transformation $T_\alpha Z := (Z^L, \dots, Z_3, \alpha^{-1}Z_2)$. Then $\alpha \mapsto \mathscr{L}(T_\alpha Z) = \sum_{l=3}^{L-1} \|\sigma_{l+1}^{-1}(Z^{l+1})Z^{l,+}\|^2 + \alpha^2 \|\sigma_3^{-1}(Z^3)Z^{2,+}\|^2$ reaches its minimum for $\alpha = 0$, which implies that $\|T_\alpha Z\| \underset{\alpha \to +\infty}{\to} +\infty$. This contradicts that $Z$ is a minimizer and shows that the infimum is reached in the infinite $Z^2$ norm limit. To avoid these situations, we study the minima of $\mathscr{L}$ subject to additional constraints.

## 4.1 Characterizing Optimal Representations: Natural Architectural Constraints

Before stating Theorem 4.2, we introduce three sets of constraints that capture essential properties of realizable representations in actual neural networks. These constraints are a way to encode properties of the underlying architecture in the space of representable intermediate embeddings. In particular, this allows one to view dually the neural network under consideration as a pure constraint on the space of intermediate representations, which forces the embedding of one layer $Z^l$ to be linked to that of the embedding before $Z^{l-1}$ through the imposed architecture. We first start with the definition of the constraints.

**Definition 4.1** (Constraints). Consider the set $\mathcal{S} = \mathcal{S}(Z^1)$ consisting of all networks $Z = (Z^2, \ldots, Z^L) \in \mathbb{R}^{n_2 \times N} \times \cdots \times \mathbb{R}^{n_L \times N}$ that satisfy the following conditions (I)-(II)-(III):

(I)  $\sigma_{l+1}^{-1}(Z^{l+1}) Z^{l,+} Z^l = \sigma_{l+1}^{-1}(Z^{l+1}), \quad \forall l = 1, \ldots, L-1;$

(II)  $s_i(Z^l) \leq C_{l+1} s_i(\sigma_{l+1}^{-1}(Z^{l+1})), \quad \forall i = 1, \ldots, r_l, \quad \forall l = 2, \ldots, L-1,$

(III)  $\mathrm{rank}(\sigma_l^{-1}(Z^l)) \geq \mathrm{rank}(Y),$

where $C_l > 0$ are arbitrary, $r_l$ is the rank of $\sigma_{l+1}^{-1}(Z^{l+1})$, $s_i(A)$ denotes the $i$-th singular value of a matrix $A$ and $\sigma_2, \ldots, \sigma_L$ is a set of entrywise omeomorphisms with $\sigma_l(x) \neq 0$ for all $x \neq 0$.

**Constraint (I): representability.** For any realizable sequence of representations $Z^1, \ldots, Z^L$, each representation $Z^{l+1}$ must satisfy $Z^{l+1} = \sigma_l(W_l Z^l)$ for some weight matrix $W_l$. When we reparameterize the representation cost functional in equation 9 in terms of the features embeddings (see equation 12), the constraint (I), by coupling the embeddings $Z^l$ and $Z^{l+1}$, expresses this representability requirement. We remark that this constraint does not restrict the optimization problem further, as it was shown in (Jacot et al., 2022) that the stationary points of equation 9 are in bijection with the ones of equation 12 with constraint (I).

**Constraint (II): effect of nonlinearity.** The second constraint (II) can be directly related to the effect of the nonlinear activations, and it represents exactly the situation for diagonal feature matrices, for which the nonlinearity acts directly on the singular values. The necessity of constraint (II) for the theoretical analysis relies on making it possible to study the alignment of the optimal basis, avoiding any possible singular value escaping to infinity. Interestingly, notice that while in (II) $C$ is an arbitrary positive constant, we observe numerically in Section 5 that a perfect fit happens for $C = 1$, reminiscent of a norm balancing condition on all the layers.

**Constraint (III): excluding lower-than rank $K$ solutions.** The third set (III) is introduced here as it arises naturally in the proof of the theorem as a consequence of (II), and it allows us to rule out the presence of global minima of lower rank. This is because, as we will show in the proof in Appendix B, all global minima satisfy the equality constraint. We notice that, as a byproduct of Theorem 4.2, we know that if we get rid of constraint (III), then all global minimizers either satisfy DNC1 or some layer embedding has rank strictly smaller than $K$. We expect these minimizers to be exceptional in some sense, as they were proven to be optimal in a very particular setting in (Súkeník et al., 2024a). We also remark that the singular values structure is also predicted recursively by Theorem 4.2, which we will numerically show also in Section 5.

We highlight here that constraint (II) corresponds to a constraint just on the nonzero singular values of the matrices $Z^l$ and $\sigma_{l+1}^{-1}(Z^{l+1})$. Therefore, (II) allows the case in which all feature matrices $Z^l$ for $l = 1, \ldots, L-1$ are full-rank. Moreover, $\mathcal{S}$ is bounded but not compact because of constraint (III), making it nontrivial to prove the existence of a minimizer. In Appendix D.1 we include a numerical experiment in which we verify if these hypotheses are satisfied in a standard training setting. The following main result on the minimizers of a constrained version of equation 12 holds:

**Theorem 4.2** (DNC1 is optimal for constrained representation cost). *Let $\sigma_2, \ldots, \sigma_L$ be a set of entrywise homeomorphisms and $\sigma_1$ an analytic non-polynomial function, with $\sigma_l(x) = 0$ if and only if $x = 0$ for every $l = 1, \ldots, L$. Assume $K = n_L \leq \cdots \leq n_1 = N \leq 2^{n_1 - 1}$. Let $\mathcal{S} \subset \mathbb{R}^{n_1 \times N} \times \cdots \times \mathbb{R}^{n_L \times N}$ be as in Definition 4.1. Then, for almost every $Z^1$, the global minima of the optimization problem*

$$\min_{Z \in \mathcal{S}} \mathscr{L}(Z)$$

*satisfy* DNC1 *for all intermediate layers $l = 1, \ldots, L$. Moreover, any global minimizer $\bar{Z}$ satisfies*

$$\bar{Z}^l \propto O_l \sigma_{l+1}^{-1}(\bar{Z}^{l+1}), \quad O_l \in \mathbb{R}^{n_l \times n_{l+1}}, \ O_l^\top O_l = I, \quad \forall l = 2, \ldots, L-1.$$

*Proof (sketch).* First, we consider the set of $Z^1 \in \mathbb{R}^{N \times N}$ such that $\mathrm{rank}(Z^1) = N$, which is a generic condition as shown in Lemma B.3. Now, fixed $Z^1$ satisfying this condition we look at the minimizers in the $Z^2$ variables, subject to the constraints in Definition 4.1. Since regularization on the second term $W_2$ is not present, there

is just a single term involving $Z^2$, leading to the following:

$$\min_{Z^2 \in \mathbb{R}^{n_2 \times N}} \|\sigma_3^{-1}(Z^3) Z^{2,+}\|^2$$
$$\text{s.t.} \quad \sigma_3^{-1}(Z^3) Z^{2,+} Z^2 = \sigma_3^{-1}(Z^3)$$
$$\sigma_2^{-1}(Z^2) Z^{1,+} Z^1 = \sigma_2^{-1}(Z^2)$$
$$s_i(Z^2) \leq C_2 s_i(\sigma_3^{-1}(Z^3)).$$

Notice that, since $Z^1$ is invertible, the second representability constraint is trivially satisfied, as $Z^{1,+} Z^1 = I_N$. Therefore, thanks to Lemma B.1, we can describe exactly the global minimizers of this optimization problem as a function of $\bar{Z}^2 \propto O_2 \sigma_3^{-1}(Z^3)$. By substituting this expression in $\mathscr{L}$, we can do recursively the same for $Z^3$, by using the third constraint on the rank of representations. $\qquad \square$

*Remark* 4.3. For almost every $Z^1$ is intended in the sense of pushforward Lebesgue measure, i.e., for almost every $(X, W_1) \in \mathbb{R}^{n_0 \times N} \times \mathbb{R}^{n_1 \times N}$.

A rigorous formulation and proof of Theorem 4.2 involves multiple technical lemmas, and can be found in Appendix B.

Theorem 4.2 can be reinterpreted as follows: if we allow the architecture to collapse already at the first layer, that is, if the kernel of $W_1$ can be large enough (i.e., $n_1 \geq N$), then, under (I)–(III), it is optimal to collapse all intermediate layers. Notice that the recursive nature of Theorem 4.2 describes all the optimal intermediate embeddings $Z^l$ recursively as a function of $\sigma_L^{-1}(Z^L)$, both in terms of singular values and singular vectors. As an example, $Z^{L-1}$ has to be proportional to a rotated version of $\sigma_L^{-1}(Y)$, and therefore we know singular values and vectors exactly. Therefore, by going backward recursively, we can recover the structure of all of the previous layers.

We emphasize that Theorem 4.2 *can be extended to residual networks under a weaker set of assumptions*. Indeed, in that case, we can get rid of all constraints, except the natural one in (I). Given the similarity of the key ideas, we present this result along with its proof in Appendix C.

In the case of feedforward networks, with a stronger request on the intermediate widths, we have the following additional main result, proving that for almost every interpolating set of parameters, there exists a continuous path in parameter space along which the loss decreases, finally converging to globally optimal points, which satisfy DNC1 for Theorem 4.2.

**Theorem 4.4** (Benign loss landscape: monotonic paths to DNC1)**.** *Assume the intermediate widths in $\mathcal{S}$ satisfy the condition $K \leq n_L = n_{L-1} = \cdots = n_1 = N$ and that the $\sigma_l$'s are analytic. Then, for almost every initial condition $Z(0) \in \mathcal{S}$, there exists a continuous path $\gamma(t)$ such that:*

1. *$\gamma(0) = Z(0)$ and $\gamma(1)$ is a global minimum of $\mathscr{L}$;*

2. *$\mathscr{L} \circ \gamma$ is continuous Lebesgue-almost everywhere (except for a finite set of times), with all discontinuities $t_1, \ldots, t_m$ removable, i.e., $\lim_{t \to t_i^-} \mathscr{L}(\gamma(t)) = \lim_{t \to t_i^+} \mathscr{L}(\gamma(t)) > \mathscr{L}(\gamma(t_i)), \quad \forall i = 1, \ldots, m;$*

3. *for all continuity times $s \geq t$ we have $\mathscr{L}(\gamma(t)) \geq \mathscr{L}(\gamma(s))$.*

*Proof (sketch).* As in Theorem 4.2, we consider the set of $Z^1 \in \mathbb{R}^{N \times N}$ such that $\text{rank}(Z^1) = N$, which is a generic condition as shown in Lemma B.3. Now, fixed $Z^1$ satisfying this condition we look at the minimizers in the $Z^2$ variables. Thanks to Lemma B.4, there exists an energy decreasing path from the initial condition to the global minimizer in $Z^2$. We recursively concatenate this with the optimal path in $Z^3$, where $Z^2$ follows the central path dynamics (staying optimal as in (Han et al., 2022)). At the end, we end up with the concatenation of $L$ loss decreasing paths in $\mathcal{S}$, which is therefore globally energy decreasing and connects the initial condition to the constrained global minimizers. $\qquad \square$

The full proof of Theorem 4.4 can be found in Appendix B.2. The main takeaway is that, from almost every interpolating initial condition (aside from a set of measure zero), one can construct a continuous path in

parameter space along which the loss decreases (up to a finite set of downward jumps) and the network converges to a global minimum characterized by deep neural collapse. In particular, Theorem 4.4 implies that

$$\inf_{\gamma \in \Gamma} \sup_{t \in [0,1]} \mathscr{L}(\gamma(t)) \leq \mathscr{L}(Z(0)),$$

where $\Gamma := \{\gamma \in \mathcal{C}^0([0,1];\mathcal{S}) \mid : \gamma(0) = Z(0), \gamma(1) \text{ satisfies } \text{DNC1} \ \forall l = 1,\ldots,L\}$.

We highlight that in any numerical method, one looks at a discretized version of the path $\gamma$, so the "probability" of seeing discontinuity points is zero, no matter how small the learning rate. In particular, no matter how small the stepsize, there always exists a sequence of points that decreases the loss and reaches DNC1 configurations. This result is exactly formalized in the following corollary, together with a more detailed explanation in Remark B.6:

**Corollary 4.5** (No loss barrier for any finite stepsize)**.** *In the setting of Theorem 4.4, for almost every $Z(0) \in \mathcal{S}$ there exists a sequence of curves $\gamma_k : [0,1] \to \mathcal{S}$ uniformly converging to the continuous curve $\gamma$ of Theorem 4.4 such that the following hold for all $k \geq 1$:*

- *$\gamma_k$ is piecewise constant in a set of disjoint intervals $\{I_{j,k} := [T_{j,k}, T_{j+1,k})\}_{j=0,\ldots,k-1}$ whose union is $[0,1]$, and with $0 < T_{j+1,k} - T_{j,k} < \frac{1}{k}$ for all $j = 0,\ldots,k-1$;*

- *$\gamma_k(0) = Z(0)$, $\gamma_k(1) \in \arg\min_{Z \in \mathcal{S}} \mathscr{L}(Z)$*

- *$\mathscr{L}$ is decreasing, i.e., $\mathscr{L}(\gamma_k(0)) \geq \mathscr{L}(\gamma_k(T_{1,k}))) \geq \mathscr{L}(\gamma_k(T_{2,k}))) \geq \cdots \geq \mathscr{L}(\gamma_k(T_{k,k}))) \geq \mathscr{L}(\gamma_k(1))$.*

The idea of the proof Corollary 4.5 relies essentially on a discretization of the path from Theorem 4.4, and can be found in Appendix B.4.

A second way to interpret Theorem 4.4 and Corollary 4.5 is that the loss landscape of $\mathscr{L}$, despite being highly nonconvex, is well behaved in the sense that there are essentially no loss barriers from almost all interpolating points to global minima. This result, being independent of the specific temporal dynamics imposed during training, provides theoretical support for the empirical "universality" of neural collapse observed in classification tasks when using sufficiently wide architectures (Rangamani et al., 2023a) and possibly gives an hint on why it is observed towards the end of training (Papyan et al., 2020; Rangamani et al., 2023a) (i.e., in the interpolating phase, where the landscape is well behaved).
Moreover, the phenomenon is "unidirectional", progressing from high to low rank as demonstrated in the proof, which is reminiscent of the behaviour observed in the deep linear case in (Wang & Jacot, 2024).

Finally, we point out that these results are recovered as the singular limit in the regime where an arbitrarily small penalization term for the first layer $\|W_2\|^2 = \|\sigma_2^{-1}(Z^2)Z^{1,+}\|^2$ is included. A natural question arises: are the highly structured solutions given by Theorem 4.2 preserved when $\|W_1\|^2$ is included? To answer this question about stability, we consider the following perturbed loss, where $\phi$ is a generic nonnegative lower-semicontinuous function defined on the set $\mathcal{S}$ (Definition 4.1),

$$\mathscr{L}_\lambda(Z) = \begin{cases} +\infty, & \text{if } Z^L \neq Y, \\ \frac{1}{2}\sum_{l=1}^{L-1} \|\sigma_{l+1}^{-1}(Z^{l+1})Z^{l,+}\|^2 + \frac{1}{\lambda}\phi(Z), & \text{otherwise,} \end{cases} \tag{13}$$

and show that, as $\lambda \to \infty$, the global minima of $\mathscr{L}_\lambda$ converge to global minima of $\mathscr{L}$. Note that varying the value of $\lambda$ is equivalent to rescaling the dataset $X$ by a factor $\lambda$.

**Theorem 4.6** (Stability of the DNC1 global minima)**.** *Assume $\phi : \mathcal{S} \to \mathbb{R}_+$ is lower semicontinuous. Then, the family of functionals $\mathscr{L}_\lambda : \mathcal{S} \to \mathbb{R}$ converge to $\mathscr{L} : \mathcal{S} \to \mathbb{R}$ as $\lambda \to +\infty$, in the $\Gamma$-convergence sense.*

Moreover, as a classical consequence of $\Gamma$-convergence, if $Z^* \in \mathcal{S}$ is an accumulation point for a sequence of minimizers $Z_n \in \arg\min_{Z \in \mathcal{S}} \mathscr{L}_{\lambda_n}(Z)$ with $\lambda_n \to +\infty$, then $\mathscr{L}(Z^*) = \min_{Z \in \mathcal{S}} \mathscr{L}(Z)$. The proof is included in Appendix B.2. We also highlight here that the existence of an accumulation point $Z^*$ is not immediate, since $\mathcal{S}$ is not compact. We finally remark that an equivalent stability result holds even for residual networks, as discussed in Remark C.3.

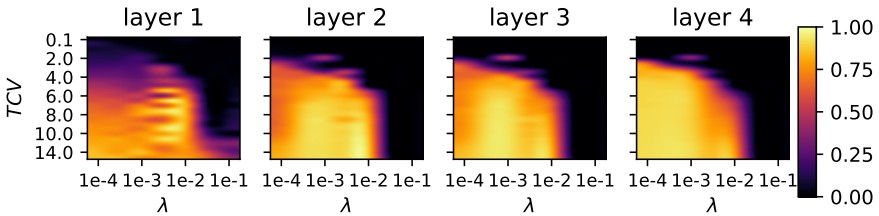

Figure 1: For each couple $(\lambda, \sigma)$ we report the relative distance $\sum_{j>K} s_j^2 / \sum_j s_j^2$ of trained weight matrices from the closest rank-$K = 10$. Here, $s_j$ are the singular values of each weight matrix.

*Remark* 4.7. Theorem 4.6 can be interpreted from two complementary perspectives. First, the optimality of DNC1 configurations established in Theorems 4.2 and C.2 remains stable under the inclusion of a backbone regularization term. Consequently, one expects the configurations across all layers, up to the first representative layer, to remain close to DNC1. In particular, such proximity to DNC1 configurations, even when the first-layer weight decay is incorporated, is expected to occur in the regime $\|X\| \gg \|Y\|$.

Alternatively, the parameter $\lambda$ in Theorem 4.6 can also be interpreted as representing the network depth. To formalize this interpretation, define

$$\mathscr{L}_L^{[l_1,l_2]}(Z) := \begin{cases} +\infty, & \text{if } Z^L \neq Y, \\ \frac{1}{L}\sum_{l=l_1}^{l_2} \phi_{l+1}(Z^{l+1}, Z^l), & \text{otherwise.} \end{cases}$$

For $Z^L = Y$, consider the difference

$$\mathscr{L}_L^{[0,L-1]} - \mathscr{L}_L^{[l^*,L-1]} = \mathscr{L}_L^{[0,l^*-1]} = \frac{1}{L}\sum_{l=0}^{l^*-1} \phi_{l+1}(Z^{l+1}, Z^l) \xrightarrow[\Gamma]{} 0, \quad \text{as } L \to +\infty.$$

Hence, in the large-depth regime, even in the absence of a pyramidal backbone structure, one expects the minimizers to exactly satisfy the DNC1 condition from the first layer $l^*$ for which the set of representable features $Z^{l^*}$ contains the minimizer predicted by Theorem 4.2, with the special case of the UFM, where the set of representable matrices $Z^{l^*}$ coincides with $\mathbb{R}^{n_{l^*} \times N}$.

This observation elucidates that, in general, for sufficiently deep architectures, the structure of the minimizers asymptotically approaches that predicted in Theorem 4.2, where $Z^2$ is effectively replaced by the minimal layer index $l^*$ such that $Z^{l^*}$ is *representative enough*.

## 5 Experimental Evaluation

### 5.1 Qualitative Behaviour of Theorem 3.5

In this section, we show in a simple setting the qualitative dependency of $\min_{Z \in \cup_{r=1}^K \mathcal{M}_r} \|W_l^* - Z\|$ on the regularization parameter $\lambda$ and on the TCV of the dataset. In particular, fixed a set of vectors $\mu_1, \ldots, \mu_K$, we consider a random Gaussian mixture dataset $X \in \mathbb{R}^{n \times N}$ with $x_i \sim N(\mu_{y_i}, \sigma^2)$, and a $Y \in \mathbb{R}^{K \times N}$ one-hot encoding matrix with $K = 10$ classes. In this way, we know that $\text{TCV}_{K,0} \sim O(\sigma^2)$, and we expect, from Theorem 3.5, the ranks to approach $K = 10$ as $\sigma^2 \to 0$. Then, for each $(\lambda, \sigma)$ in a grid of values, we sampled $X, Y$ according to the distribution described above, and we trained a four-layer fully connected neural network with the loss defined in Equation (3). In Figure 1 we report the results of this experiment. We can observe that consistently, for each $\lambda$ in our grid, we have that all matrices are essentially of rank $K$ for $\sigma$ small enough. Moreover, we also observe the decreasing behavior with respect to $\lambda$ consistently with Theorem 3.5.

### 5.2 Convergence to the Optimal Solution of Theorem 4.4

We start by illustrating and numerically validating what we claimed in Theorems 4.4 and 4.6. In particular, by setting $\delta = 1/\lambda$ in equation 13 we can show that for $\delta \to 0$, the singular values of all the intermediate

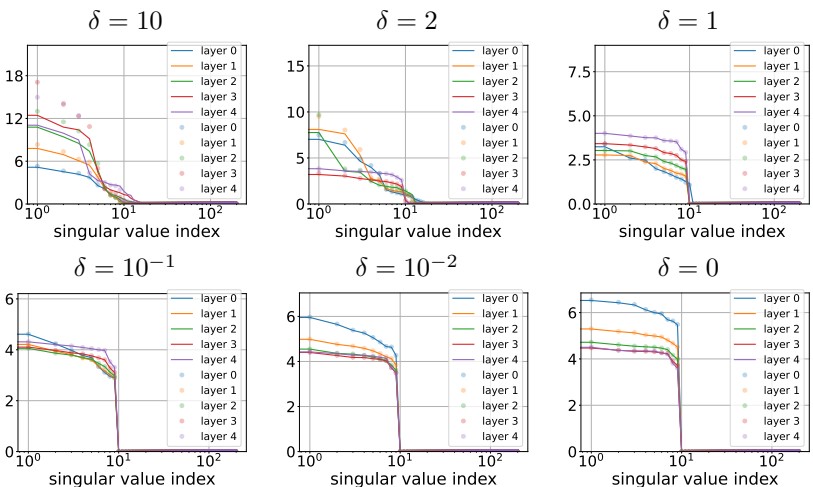

Figure 2: Singular values of trained networks (full line) against those predicted by Theorem 4.2 (dots) for different values of $\delta = \frac{1}{\lambda} \in \{10, 2, 1, 0.1, 0.01, 0\}$. The singular values are converging to the predicted ones as $\delta \to 0$, as shown by Theorem 4.6 in combination with Theorem 4.2.

post-activation features are the ones described in Theorem 4.2. To set up the experiment, we subsampled the MNIST dataset (Deng, 2012) with $N = 200$ examples, and we considered a 5 layer feedforward network with all $\sigma_l = \tanh$ activations and widths equal to $n_1 = \cdots = n_4 = N = 200, n_5 = K = 10$. By including $\delta$, the loss becomes $\mathcal{L}_{\lambda,\delta} = \frac{1}{2}\|f_\Theta(X) - Y\|^2 + \frac{\lambda}{2}\sum_{l=2}^{L}\|W_l\|^2 + \frac{\lambda\delta}{2}\|W_1\|^2 + \frac{\lambda'}{2}\sum_{l=1}^{L}\|Z^l\|^2$, where the last regularization term can be thought of as a weaker way of imposing the second set of constraints in Theorem 4.4. In particular, for this experiment we considered $\lambda = \lambda' = 10^{-5}$.

We trained the network until convergence (way past interpolation, with a final data loss lower than $10^{-6}$ in all examples) for different values of $\delta$, and we compared the actual singular values of the feature matrices after training with the ones predicted by Theorem 4.2. The results are shown in Figure 2. As we can notice, the singular values are converging to the ones predicted by Theorem 4.2, and the convergence with respect to $\delta$ seems to be fast. In fact, already for moderate sizes of $\delta$ (e.g., $\delta = 2$), the singular values are close to the predicted ones.

### 5.3 Convergence to DNC1 Configurations

In this experiment, we will showcase the convergence of TCV to zero of all intermediate layers in the setting of Theorem 4.2. The experimental setting is the same as in Section 5.2, but with $L = 10$ layers and $\delta = 0$ fixed. In Figure 3 (rightmost figure), we plot for a single run the TCV of all intermediate post-activation embeddings. As we can observe from this first plot, the TCV of all intermediate embeddings goes essentially to machine precision as expected from Theorems 4.2 and 4.4. In Figure 3 (first three figures from the left), we instead plot the average TCV across all intermediate layers, for 100 different random unitary initializations, to avoid slow convergence caused by the depth. As we can see, the TCV consistently goes to zero in all runs. Regarding the numerical ranks, we observe that they are all close to the optimal ones ($K = 10$).

## 6 Conclusions

We proposed a theoretical investigation on the emergence of low-rank weight matrices and deep-neural collapse for a general class of neural networks. In particular, we established a quantitative relationship between deep neural collapse and the rank collapse of the weight matrices. In addition, we proved that for a general class of models, by not regularizing the first layer, DNC1 is optimal for all intermediate representations under a set of natural constraints, and that essentially there is no loss barrier between the interpolating global minima and DNC1 configurations. Lastly, we showed that the DNC1 configurations are stable and are preserved even when the first weight decay is included. Future investigations will go in the direction of relaxing some

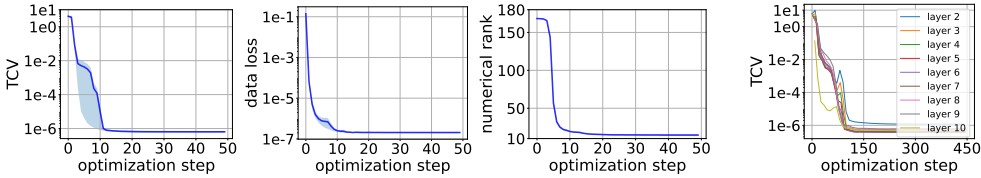

Figure 3: First three plots from the left: average of different quantities on all intermediate layers and for 100 different random initializations. Shadows represent the area included between the minimal and maximal values observed. Each optimization step represents 100 epochs. Right: TCV of every layer during training.

constraints and trying to adapt this approach to attention-based networks and GNNs. Moreover, we expect the same tools to be of interest when studying the representation learning of DNNs in general, which is also planned for future investigation.

**Acknowledgements**

SB would like to thank the Gran Sasso Science Institute for supporting his research visits in 2023 and 2025, and acknowledges support from the Natural Sciences and Engineering Research Council of Canada and Fonds de recherche du Qébec - Nature et Technologies through grants RGPIN-2020-06766 and 359708, respectively. The work of FT is partially funded by the PRIN-MUR project MOLE code: 2022ZK5ME7 and by the PRIN-PNRR project FIN4GEO within the European Union's Next Generation EU framework, Mission 4, Component 2, CUP P2022BNB97. The work of E. Zangrando was funded by the MUR-PNRR project "Low-parametric machine learning". P. Deidda was supported by the MUR-PRO3 grant "STANDS" and the INdAM-GNCS grant "NLA4ML—Numerical Linear Algebra Techniques for Machine Learning".

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

## A  Useful Definitions and Results

In this section, we will introduce some useful definitions and fact which we will need to prove the results in Appendix B. We start recalling the definition and some basic facts about $\Gamma$-convergence of functionals (see (Dal Maso, 1993)).

**Definition A.1.** ($\Gamma$-convergence)  Let $S$ be a locally numerable topological space and let $F_n : S \to \mathbb{R}$ be a sequence of functionals and $F : S \to \mathbb{R}$ be another functional. $F_n$ is said to $\Gamma$-converge to $F$ if the following hold:

(i) For any sequence $(x_n) \subset S$ with $x_n \to x$, we have

$$F(x) \leq \liminf_{n \to \infty} F_n(x_n).$$

(ii) (Recovery sequence condition). For any $x \in S$, there exists a sequence $x_n \to x$ such that

$$F(x) \geq \limsup_{n \to \infty} F_n(x_n).$$

To build some intuition, the first condition means that asymptotically, $F$ is a common lower bound for the sequence of $F_n$, at least pointwise. The second condition is a condition of recovery of the lower bound, implying, in a certain sense, optimality of the lower bound. This definition can be useful when it is simpler to characterize the minima of $F_n$ than those of $F$, or viceversa. In our case, we can characterize the minima of $F$, but not those of $F_n$. However, thanks to $\Gamma$-convergence, we can establish that the minima of $F_n$ are going be close to those of $F$ in suitable scenarios. In particular, the following well known result holds:

**Proposition A.2.** *(Dal Maso, 1993) Let $F_n \to F$ in the sense of $\Gamma$-convergence and let $S$ be sequentially compact. Then, for every sequence $x_n \in \arg\min_{y \in S} F_n(y)$ and every limit point $x$ of $(x_n)$, we have*

$$x \in \arg\min_y F(y).$$

The next lemma discusses the zeros of single-variable analytic functions, showing that their cardinality is always finite on compact sets, which we will need later. The results is well known and can be found in (Krantz & Parks, 1992).

**Lemma A.3** (Finite number of zero of real analytic functions on compact intervals)**.** *Consider a nonzero real analytic function $f : [0, 1] \to \mathbb{R}$. Then $f$ has a finite number of zeros in $[0, 1]$.*

*Proof.* First, we show that zeros are isolated. Let $t_0 \in [0, 1]$ be a zero of $f$. Since $f$ is real analytic, then we can write

$$f(t) = \sum_{k=0}^{+\infty} a_k (t - t_0)^k, \quad |t - t_0| \leq \rho$$

since $f \neq 0$, there exists $m$ such that $a_m \neq 0$. Therefore,

$$f(t) = (t - t_0)^m h(t), \quad h(t) := \sum_{k=0}^{+\infty} a_{m+k}(t - t_0)^k, \quad h(t_0) = a_m \neq 0$$

Therefore, by continuity of $h$ there exists a neighborhood of $t_0$ of radius $\rho'$ such that $h(t) \neq 0$ for all $|t - t_0| < \rho'$. Then in this interval we have $f(t) = 0 \iff t = t_0$, proving that $t_0$ is isolated. Now, assume $f$ has infinitely many zeros $(t_k)$ in $[0, 1]$, then by compactness there exists a convergent subsequence $t_{k_j} \to \bar{t} \in [0, 1]$. But then, $\bar{t}$ would be an accumulation point, and this is only possible if $f$ is zero in an interval around $\bar{t}$. Since $f$ is analytic, this would force $f \equiv 0$, which is a contradiction. $\square$

**Proposition A.4** (Lower semicontinuity of functions defined through infimum)**.** *Let $X, Y$ be two finite-dimensional normed spaces, $f : X \times Y \to \mathbb{R}$ and $g : X \times Y \to \mathbb{R}^m$ be continuous functions, and $\Omega(x) = \{y \in$*

$Y \mid g(x, y) = 0\}$. *Moreover, assume that $f(x, y)$ is coercive with respect to $y$, uniformly in $x$, i.e. that there exists $C > 0$ such that $\|f(x, y)\| \geq C\|y\|$ for all $x \in X$. Then, the function*

$$F(x) = \inf_{y \in \Omega(x)} f(x, y)$$

*is lower semicontinuous.*

*Proof.* To prove the lower semicontinuity of $F(x)$ we need to show that, for every sequence $x_k \to \bar{x}$, we have

$$\liminf_k F(x_k) \geq F(\bar{x}).$$

Let $(x_{k_j})$ be a subsequence of $(x_k)$ such that $\exists \lim_j F(x_{k_j}) = \liminf_k F(x_k)$ Then by definition of infimum, for every $x_k$ we can find $y_k \in \Omega(x_k)$ such that $F(x_k) \leq f(x_k, y_k) \leq F(x_k) + \frac{1}{k}$. Then we have two cases:

(i) If the sequence $(y_{k_j})$ is unbounded, up to considering a subsequence, we can assume that $\|y_{k_j}\| \to \infty$. Then, for the uniform coercivity of $f$ we have

$$f(x_{k_j}, y_{k_j}) \to \infty$$

and, therefore,

$$\liminf_k F(x_k) = \lim_j F(x_{k_j}) \geq \lim_j \left( f(x_{k_j}, y_{k_j}) - \frac{1}{k_j} \right) = +\infty \geq F(\bar{x}).$$

(ii) If the sequence $(y_{k_j})$ is bounded, up to considering a subsequence, we can assume $y_{k_j} \to \bar{y}$. Since $y_{k_j} \in \Omega(x_{k_j})$ for all $j$, by the continuity of $g$, $0 \equiv g(x_{k_j}, y_{k_j}) \to g(\bar{x}, \bar{y})$. Therefore, $\bar{y} \in \Omega(\bar{x})$. Similarly, the continuity of $f$ gives

$$f(x_{k_j}, y_{k_j}) \to f(\bar{x}, \bar{y}).$$

Therefore, we see that

$$\liminf_k F(x_k) = \lim_j F(x_{k_j}) \geq \lim_j \left( f(x_{k_j}, y_{k_j}) - \frac{1}{k_j} \right) = f(\bar{x}, \bar{y}) \geq \inf_{y \in \Omega(\bar{x})} f(\bar{x}, y) = F(\bar{x}),$$

which is the required inequality.

This concludes the proof. $\qquad\square$

*Remark* A.5. We will use this last result in Appendix B.2 to prove the lower semicontinuity of $F(A, B) = \|AB^+\|^2$. In fact, by defining

$$x = (A, B) \in X = \{(A, B) \in \mathbb{R}^{n \times N} \times \mathbb{R}^{m \times N} \mid AB^+B = A\} \quad \text{and} \quad y = W \in \mathbb{R}^{n \times m}$$

we can define $g(x, y) = WB - A$, $f(x, y) = \|W\|^2$. Then, recalling that $\Omega(x) = \{y = W \mid g(x, y) = WB - A = 0\}$, a standard result, see e.g. (Golub & Van Loan, 2013, Section 5.5.1), yields

$$F(x) = \inf_{y \in \Omega(x)} f(x, y).$$

In particular we are exactly in the setting of Proposition A.4, proving lower semicontinuity of $F(A, B) = \|AB^+\|^2$.

**Lemma A.6** (Blowup of $\mathscr{L}$ outside the natural constraint)**.** *Consider a fixed matrix $A \in \mathbb{R}^{n \times N}$ with $N \geq n \geq \mathrm{rank}(A) = r$ and the matrix functions $f, g : \mathbb{R}^{m \times N} \to \mathbb{R}$ defined as $g(B) = \|AB^+B - A\|^2$ and $f(B) = \|AB^+\|$ with $m \geq n$. Let $B_k \to \bar{B}$ with $g(B_k) = 0$ for every $k$ and $g(\bar{B}) \neq 0$. Then, $\lim_k f(B_k) = +\infty$.*

*Proof.* To prove it, let $U_k \Sigma_k V_k^T$ be the reduced SVD decomposition of $B_k$ and $\bar{U}\bar{\Sigma}\bar{V}^T$ the one of $\bar{B}$. Then by assumption we have $A = A\bar{V}\bar{V}^T + A(I - \bar{V}\bar{V}^T)$ with the second term on the right hand side being different from zero because $g(\bar{B}) \neq 0$. Additionally, w.l.o.g., for any $k$ we can write $V_k = [V_{k,1}, V_{k,2}]$, $\Sigma_k = \text{diag}[\Sigma_{k,1}, \Sigma_{k,2}]$ and $U_k = [U_{k,1}, U_{k,2}]$ where $[U_{k,1}, \Sigma_{k,1}, V_{k,1}] \to [\bar{U}, \bar{\Sigma}, \bar{V}]$, $I - V_{k,1}V_{k,1}^T \to (I - \bar{V}\bar{V}^T)$ and $\Sigma_{k,2} \to 0$. Then

$$f(B_k) = \|AV_k^1\Sigma_{k,1}^{-1}U_{k,1}^T + AV_{k,2}\Sigma_{k,2}^{-1}U_{k,2}^T\| \geq \|AV_{k,2}\Sigma_{k,2}^{-1}\| - \|AV_{k,1}\Sigma_{k,1}^{-1}\|$$

$$\geq \min_i (\Sigma_{k,2})_{ii}^{-1}\|AV_{k,2}\| - \|AV_{k,1}\Sigma_{k,1}^{-1}\| \to +\infty - \|A\bar{V}\bar{\Sigma}^{-1}\| = +\infty,$$

where we have used that the singular values are all nonnegative, i.e., $\Sigma_{k_{ii}} \geq 0$ for all $i$ and $k$ with $\Sigma_{k,2_{ii}} \to 0$ for any i, that $\lim_k AV_{k,2} = A(I - \bar{V}) \neq 0$ and the equalities $\|AV_{k,1}\Sigma_{k,1}^{-1}U_{k,1}^T\|^2 = Tr(AV_{k,1}\Sigma_{k,1}^{-1}U_{k,1}^T U_{k,1}\Sigma_{k,1}^{-1}V_{k,1}^T A) = Tr(AV_{k,1}\Sigma_{k,1}^{-1}\Sigma_{k,1}^{-1}V_{k,1}^T A) = \|AV_{k,1}\Sigma_{k,1}^{-1}\|^2$ and analogously $\|AV_{k,2}\Sigma_{k,2}^{-1}U_{k,2}^T\| = \|AV_{k,2}\Sigma_{k,2}^{-1}\|$. $\square$

## A.1 Proof of Proposition 3.1

The main step of the proof is to show that the gradient with respect to a single data point produces matrices of rank at most 1. One then concludes by observing that the sum of $K$ terms of rank at most 1 has rank at most $K$. Given the structure of the loss function that depends only on the output of the model $f(\Theta, x)$, if we use the chain rule and the recursive definition of $f(\Theta, x)$ in equation 1, we obtain

$$\nabla_{W_l} \ell\left(f(\Theta, x), y\right) = \nabla_{W_l} \ell\left(f_l(\Theta_l, z^l(x)), y\right) = \frac{\partial f(\Theta, x)}{\partial W_l}^\top \nabla_f \ell\left(f(\Theta, x), y\right)$$

$$= \frac{\partial z^l(x)}{\partial W_l}^\top \frac{\partial f_l(\Theta_l, z^l(x))}{\partial z^l}^\top \nabla_f \ell\left(f(\Theta, x), y\right).$$

Now we use the assumption that $z^l = \sigma(W_l z^{l-1}) = \sigma(a^l)$, where we have introduced the auxiliary variable $a^l := W_l z^{l-1}$. This finally leads to

$$\nabla_{W_l} \ell\left(f(\Theta, x), y\right) = \left(z^{l-1}(x) \otimes I\right)\frac{\partial z^L(x)}{\partial a^l}^\top \nabla_{z^L} \ell\left(z^L(x), y\right)$$

$$= \frac{\partial f_l(\Theta_l, \sigma(a^l(x)))}{\partial a^l}^\top \left(\nabla_f \ell\left(f(\Theta, x), y\right) \cdot z^{l-1}(x)^\top\right).$$

By noticing that the last term $\nabla_f \ell\left(f(\Theta, x), y\right) z^{l-1}(x)^\top$ is a matrix of rank at most 1, we can conclude by submultiplicativity of the rank. $\square$

## A.2 Proof of Lemma 3.4

For simplicity, we will omit the superscript $j$ from $z_i^j$. We start by considering the Taylor expansion of the gradient $\nabla_{W_l} \ell\left(f_j(\Theta_j, z), y\right)$ centered in $(\bar{z}_k, \bar{y}_k)$ for any point $(z_i, y_i)$ with $i \in \mathcal{C}_k$ and any $k = 1, \ldots, K$

$$\nabla_{W_l} \ell\left(f_j(\Theta_j, z_i), y_i\right) = \nabla_{W_l} \ell\left(f_j(\Theta_j, \bar{z}_k), \bar{y}_k\right)$$
$$+ \nabla_{(z,y)} \nabla_{W_l} \ell\left(f_j(\Theta_j, \bar{z}_k), \bar{y}_k\right)\left[(z_i, y_i) - (\bar{z}_k, \bar{y}_k)\right] + R_i^k(\Theta),$$

where the reminder is given by

$$R_i^k(\Theta) := \frac{1}{2}\nabla_{(z,y)}^2 \nabla_{W_l} \ell\left(f_j(\Theta_j, \alpha_i^k)\beta_i^k\right)[(z_i, y_i) - (\bar{z}_k, \bar{y}_k), (z_i, y_i) - (\bar{z}_k, \bar{y}_k)],$$

for some $(\alpha_i^k, \beta_i^k)$ between $(z_i, y_i)$ and $(\bar{z}_k, \bar{y}_k)$. Therefore, using the linearity of $\nabla_{(z,y)} \nabla_{W_l} \ell[\cdot]$, we have

$$
\begin{aligned}
\nabla_{W_l} \mathcal{L}_0^j(\Theta) = & \frac{1}{N} \sum_{j=1}^{N} \nabla_{W_l} \ell \left( f_j(\Theta_j, z_j), y_j \right) = \\
= & \frac{1}{N} \sum_{k=1}^{K} \sum_{i \in \mathcal{C}_k} \nabla_{W_l} \ell \left( f_j(\Theta_j, z_i), y_i \right) = \\
= & \frac{1}{N} \sum_{k=1}^{K} \sum_{i \in \mathcal{C}_k} \nabla_{W_l} \ell \left( f_j(\Theta_j, \bar{z}_k), \bar{y}_k \right) + \\
& + \nabla_{(z,y)} \nabla_{W_l} \ell \left( f_j(\Theta_j, \bar{z}_k), \bar{y}_k \right) [(z_i, y_i) - (\bar{z}_k, \bar{y}_k)] + R_i^k(\Theta) = \\
= & \sum_{k=1}^{K} \pi_k \nabla_{W_l} \ell \left( f_j(\Theta_j, \bar{z}_k), \bar{y}_k \right) + \\
& + \frac{1}{N} \sum_{k=1}^{K} \nabla_{(z,y)} \nabla_{W_l} \ell \left( f_j(\Theta_j, \bar{z}_k), \bar{y}_k \right) \underbrace{\left[ \sum_{i \in \mathcal{C}_k} ((z_i, y_i) - (\bar{z}_k, \bar{y}_k)) \right]}_{=0} + \\
& + \frac{1}{N} \sum_{k=1}^{K} \sum_{i \in \mathcal{C}_k} R_i^k(\Theta) =: \nabla_{W_l} \mathcal{L}_0^{j,\mathcal{C}}(\Theta) + R(\Theta),
\end{aligned}
$$

where $\pi_k = N_k/N$. Hence, we obtain

$$
\|R(\Theta)\| \leq \frac{1}{N} \sum_{k=1}^{K} \sum_{i \in \mathcal{C}_k} \|R_i^k(\Theta)\| \leq \frac{M_{\ell,j}(\Theta)}{2N} \sum_{k=1}^{K} \sum_{i \in \mathcal{C}_k} \|(z_i^k, y_i^k) - (\bar{z}_k, \bar{y}_k)\|^2 = \frac{M_{\ell,j}(\Theta)}{2} \, \text{WCSS}_j,
$$

where

$$
\begin{aligned}
M_{\ell,j}(\Theta) = & \sup_{k=1,\ldots,K} \sup_{i \in \mathcal{C}_k} \| \nabla_{(z,y)}^2 \nabla_{W_l} \ell \left( f_j(\Theta_j, \alpha_i^k), \beta_i^k \right) \| \leq \\
& \leq \sup_{k=1,\ldots,K} \sup_{(\alpha,\beta) \in \mathcal{H}_k^j} \| \nabla_{(z,y)}^2 \nabla_{W_l} \ell \left( f_j(\Theta_j, \alpha), \beta \right) \|,
\end{aligned}
$$

as desired. $\qquad\square$

## A.3 Proof of Theorem 3.5

Observe that from equation 4, at the stationary point we have

$$
W_l^* = -\frac{1}{\lambda} \nabla_{W_l} \mathcal{L}_0(\Theta^*). \tag{14}
$$

Now, Lemma 3.4 allows us to control the distance of the gradient in the right-hand side of equation 14 from $\mathcal{M}_K^{n_{l-1},n_l}$. In particular, given $\mathcal{C} \in \cup_{r=1}^{K} \mathcal{P}_r$, for any $j = 1, \ldots, l-1$ we have

$$
\begin{aligned}
\min_{Z \in \cup_{r=1}^{K} \mathcal{M}_r} \|W_l^* - Z\| & \leq \frac{\| \nabla_{W_l} \mathcal{L}_0(\Theta^*) - \nabla_{W_l} \mathcal{L}_0^{j,\mathcal{C}}(\Theta^*) \|}{\lambda} \\
& \leq \frac{M_{l,j}(\Theta^*, \mathcal{C}) \, \text{WCSS}_j(\mathcal{C})}{\lambda},
\end{aligned}
$$

where we have used that $\nabla_{W_l} \mathcal{L}_0^{j,\mathcal{C}}(\Theta^*)/\lambda$ has rank smaller than or equal to the number of families of $\mathcal{C}$. Then, first minimizing over $j = 1, \ldots, l-1$, second bounding from above $M_{l,j}(\Theta^*, \mathcal{C}) \leq M_{l,j}(\Theta^*)$ for any $j$, and third minimizing over all the possible partitions $\mathcal{C} \in \cup_{r=1}^{K} \mathcal{P}_r$ yield the thesis.

# B  Global Minima of Representation Cost for Feedforward Networks

In this section, we will prove the results presented in Section 4 together with the necessary lemmas. As we already discussed in Section 4, the main idea is to study the minima of $\mathscr{L}$ recursively layer by layer, i.e., to study the minima of the local terms $\mathscr{L}_{loc}(Z^{l+1}, Z^l) = \frac{1}{2}\|\sigma_{l+1}^{-1}(Z^{l+1})Z^{l,+}\|^2$ in $Z^l$ for $Z^{l+1}$ fixed. We characterize such minimum, under the constraints considered in Theorem 4.2, in the next technical lemma. This will serve as a basis for recursively proving the main Theorem 4.2. We start formally introducing the constraints. Consider $A \in \mathbb{R}^{n \times N}$ a fixed matrix playing the role of $\sigma_{l+1}^{-1}(Z^{l+1})$, with $\mathrm{rank}(A) = r$, and $\{s_i(A)\}$ its singular values. Then, for some fixed $m \geq n$ and $C > 0$ we write

$$\mathcal{S}_{loc}(A) := \Big\{ B \in \mathbb{R}^{m \times N} : s_i(B)^2 \leq C s_i(A)^2 \ \forall i = 1, \ldots, r, \ AB^+B = A \Big\}. \tag{15}$$

**Lemma B.1** (Optimal single layer for feedforward representation cost). *Let $A \in \mathbb{R}^{n \times N}$ be a fixed matrix with $\mathrm{rank}(A) = r$, and $\{s_i(A)\}$ its singular values and let also $\mathscr{L}_{loc}(B) = \frac{1}{2}\|AB^+\|^2$. Then, for $n \leq m$ and any fixed constant $C > 0$ in the definition of $\mathcal{S}_{loc}(A)$, we have that the solution to the constrained optimization problem $\min_{B \in \mathcal{S}_{loc}(A)} \mathscr{L}_{loc}(B)$ is attained at all points*

$$B^* = OA, \quad O \in \mathbb{R}^{m \times n}, \quad O^\top O = I.$$

*In particular, for any optimal $B^*$, we have $\mathscr{L}(B^*) = \mathrm{rank}(A)/2C$. Moreover, for $n = m = N$, and for Lebesgue-almost every $B_0 \in \mathcal{S}$, there exists a continuous path $\gamma : [0,1] \to \mathbb{R}^{N \times N}$ with $\gamma(0) = B_0$, $\gamma(1) \in \underset{B \in \mathcal{S}_{loc}(A)}{\arg\min} \mathscr{L}_{loc}(B)$ and $\frac{d}{dt}\mathscr{L}_{loc}(\gamma(t)) \leq 0$.*

*Proof.* First of all, we prove the existence of a minimizer. Observe that whenever a sequence $B_k \to \bar{B}$ with $A\bar{B}^+\bar{B} \neq A$, then $\mathscr{L}_{loc}(B_k) \to +\infty$ thanks to Lemma A.6. Therefore, we can restrict our attention to sequences $B_k \to \bar{B}$ with $A\bar{B}^+\bar{B} = A$ (and therefore in particular with $\mathrm{rank}(\bar{B}) \geq \mathrm{rank}(A)$).

Since the constraint set $\mathcal{S}_{loc}(A)$ is bounded, any sequence $B_k$ for which $\mathscr{L}_{loc}(B_k) \to \inf_{B \in \mathcal{S}} \mathscr{L}_{loc}(B)$ is also bounded. Therefore, there exists a convergent subsequence $B_{k_j} \to \bar{B}$ with $\mathrm{rank}(\bar{B}) \geq \mathrm{rank}(A)$. By lower semicontinuity of $\mathscr{L}_{loc}$ (shown in Remark A.5) we obtain

$$\lim_j \mathscr{L}_{loc}(B_{k_j}) = \liminf_j \mathscr{L}_{loc}(B_{k_j}) = \inf_{B \in \mathcal{S}} \mathscr{L}_{loc}(B) \geq \mathscr{L}_{loc}(\bar{B}).$$

Since the singular values $s_i(B)$ are continuous functions of $B$, the inequality $s_i(B)^2 \leq C s_i(A)^2$ gives a closed set, and therefore $s_i(\bar{B})^2 \leq C s_i(A)^2$ . Moreover, we restricted without loss of generality to sequences $(B_k)$ for which $A\bar{B}^+\bar{B} = A$ and therefore we have $\bar{B} \in S_{loc}$. Combining $\bar{B} \in S_{loc}$ with $\inf_{B \in \mathcal{S}} \mathscr{L}_{loc}(B) \geq \mathscr{L}_{loc}(\bar{B})$ we get that $\bar{B}$ is a minimizer of $\mathscr{L}_{loc}$ in $\mathcal{S}_{loc}(A)$.

To understand the structure of minimizers, decompose $A, B$ using a reduced SVD to get $A = U_A \Sigma_A V_A^\top$, $B = U_B \Sigma_B V_B^\top$. Notice that the constraint $AB^+B = A$ implies that $\mathrm{rank}(A) \leq \mathrm{rank}(B)$. Then the constraint is satisfied if and only if

$$U_A \Sigma_A V_A^\top V_B \Sigma_B^+ U_B^\top U_B \Sigma_B V_B^\top = U_A \Sigma_A V_A^\top V_B V_B^\top = U_A \Sigma_A V_A^\top.$$

Multiplying on the left by $\Sigma_A^+ U_A^\top$ we obtain

$$V_A^\top V_B V_B^\top = V_A^\top$$

and by multiplying on the right by $V_A$ we get

$$\underbrace{(V_A^\top V_B)}_{=:Q}(V_B^\top V_A) = QQ^\top = I.$$

Concerning the function value, by substituting $V_A^\top V_B = Q$ we see that

$$
\begin{aligned}
\mathscr{L}_{loc}(U_B \Sigma_B V_B) &= \frac{1}{2}\|U_A \Sigma_A V_A^\top V_B \Sigma_B^+ U_B^\top\|^2 = \frac{1}{2}\|U_A \Sigma_A Q \Sigma_B^+ U_B^\top\|^2 = \\
&= \frac{1}{2}\mathrm{tr}(U_A \Sigma_A Q \Sigma_B^+ U_B^\top U_B (\Sigma_B^+)^\top Q^\top \Sigma_A^\top U_A^\top) = \\
&= \frac{1}{2}\mathrm{tr}(\Sigma_A Q \Sigma_B^+ (\Sigma_B^+)^\top Q^\top \Sigma_A^\top) = \frac{1}{2}\|\Sigma_A Q \Sigma_B^+\|^2 = \\
&= \frac{1}{2}\sum_{i=1}^{r}\sum_{j=1}^{m} s_i(A)^2 s_j(B)^{-2} Q_{ij}^2 = \frac{1}{2}\sum_{j=1}^{m} s_j(B)^{-2}\sum_{i=1}^{r} s_i(A)^2 Q_{ij}^2.
\end{aligned}
\tag{16}
$$

Assume without loss of generality that the singular values of both $A, B$ are ordered in a descending manner. Now, since $QQ^\top = I$, this is a classical form of Brockett's cost function (Brockett, 1991), whose solution is given by $Q^* = [I_r, 0_{r,m-r}]$ by using Theorem 2.1 in (Liang et al., 2023), which is a generalization of the classical Ky-Fan trace inequality. Then, we have

$$
\min_{s_j(B)>0} \frac{1}{2}\sum_{j=1}^{r}\frac{s_j(A)^2}{s_j(B)^2}, \quad \text{s.t. } s_j(B)^2 \le C s_j(A)^2, \forall j = 1, \ldots, r.
$$

which is minimized by $s_k(B)^2 = C s_k(A)^2$. The optimal function value is given by

$$
\mathscr{L}_{loc}(U_B \Sigma_A^* V_A^\top) = \frac{1}{2}\sum_{j}\frac{s_j(A)^2}{C s_j(A)^2} = \frac{\mathrm{rank}(A)}{2C}.
$$

Therefore, we have that minima are attained with $V_B(Q^*)^\top = V_A$ for $Q^* = [I_r, 0_{r,m-r}] \in \mathbb{R}^{r \times m}$, $\Sigma_B \propto$ blockdiag$(\Sigma_A, 0_{m-r})$. This implies $V_B(Q^*)^\top Q^* = V_A Q^*$. Therefore, the $r$ dominant right eigenvectors of $B$ coincide with those of $A$. Since the last $m-r$ singular values of $B$ are zero, we can use the reduced SVD form and delete the last $m-r$ columns of $V_B$, therefore obtaining $V_B = V_A$ and $\Sigma_B \propto \Sigma_A$. Moreover, $U_B$ is free, therefore all the optimal points are

$$
B^* \propto O U_A \Sigma_A V_A^\top = OA, \quad O \in \mathbb{R}^{m \times n}, O^\top O = I \text{ (since } n \le m)
$$

and the optimal value is given by

$$
\mathscr{L}_{loc}(B^*) = \frac{\mathrm{rank}(A)}{2C}.
$$

Now, to prove the existence of the path $\gamma(t)$, we assume $n = m = N$ and, without loss of generality, we consider $C = 1$. From Equation (16) it is sufficient to find a path for the singular values $(s_i(t))$, and for the matrix $Q(t) \in O_N(\mathbb{R})$ such that they make $\mathscr{L}_{loc}$ decrease and reach the optimal value.

The existence of a descent path in $Q$, for fixed $\Sigma_A, \Sigma_B^+$, is guaranteed by Lemma B.4. Then, we can assume w.l.o.g. that $Q = I$ by assuming that the singular values of $A$ and $B$ are both ascendingly ordered. So we concatenate the path in $Q$ with the following one for the singular values of $B$:

$$
s_i(B)(t) = (1-t)s_i(B_0) + t s_i(A).
\tag{17}
$$

The path in $Q$ decreases $\mathscr{L}$ by Lemma B.4. Additionally, by taking the derivative in $t$ of $\mathscr{L}_{loc}$ along the path equation 17, from equation 16 we have

$$
\begin{aligned}
\frac{d}{dt}\mathscr{L}_{loc}(O U_A \Sigma_B(t) Q(1)^\top V_A^\top) &= \frac{d}{dt}\sum_{i,j=1}^{N} s_i^2(A) s_j^{-2}(B)(t) Q(1)_{ij}^2 \\
&= -2\sum_{i,j=1}^{N} s_i(A)^2 s_j(B)(t)^{-3}\delta_{ij}(s_j(A) - s_j(B)(0)) \\
&= -2\sum_{i=1}^{N} s_i^2(A) s_i(B)(t)^{-3}(s_i(A) - s_i(B)(0)) \le 0,
\end{aligned}
$$

where in the second equality we have used that $Q(1) = I$ and the last inequality holds because $s_i(B_0) \leq s_i(A)$. This concludes the proof. $\qquad\square$

*Remark* B.2 (Local lemma for backbone). In the case of a deep-architecture, the second layer requires a dedicated analysis because of the double representation constraint with the first layer

$$\min_{B \in \mathcal{S}_{loc}(A),\, \sigma_2^{-1}(B)Z^{1,+}Z^1 = \sigma_2^{-1}(B)} \mathscr{L}_{loc}(B).$$

In particular, if the columns of $Z^1$ are linearly independent, then the second representation constraint is trivially satisfied. In fact, in that case $Z^{1,+}Z^1 = Z^{1,-1}Z^1 = I_N$, reducing to the case presented in Lemma B.1. Thanks to this observation and Lemma B.3, the following analysis holds in the pyramidal architecture hypothesis and with a fixed $W_1$ which satisfies the assumptions of Lemma B.3.

**Lemma B.3** (Generic linear independence after a wide layer). *Consider an analytic non-polynomial activation function $\sigma : \mathbb{R} \to \mathbb{R}$ and a set of data points $\{x_i\}_{i=1}^N \subset \mathbb{R}^n$ with $n \leq N \leq 2^{n-1}$. Then for, Lebesgue-almost every couple $(X, W) \in \mathbb{R}^{n \times N} \times \mathbb{R}^{N \times n}$ it holds that $\det(\sigma(WX)) \neq 0$, where $X = [x_1, \ldots, x_N] \in \mathbb{R}^{n \times N}$ and $\sigma$ applies entrywise.*

*Proof.* Letting $Z(W, X) := WX$, consider the determinant

$$\det(\sigma(WX)) =: F(Z(W, X)),$$

which is analytic in $(W, X)$. Therefore, if we can prove that $F$ is not identically zero, then its zero set in $\mathbb{R}^{N \times n} \times \mathbb{R}^{n \times N}$ has Lebesgue measure zero (Mityagin, 2020, Proposition 1), and we can conclude. To show this, we need to exhibit a single matrix $Z$ with $\text{rank}(Z) \leq n$ for which $F(Z) \neq 0$. Following (Chu et al., 2024, Theorem 2.5), whenever $\sigma$ is non-polynomial, then $\sigma$ is not rank preserving ( see (Chu et al., 2024, Definition 2.2)), meaning that there exists a matrix $Z$ with $\text{rank}(Z) \leq \log_2(N) + 1 \leq n$ such that $\text{rank}(\sigma(Z)) = N$. This concludes the proof. $\qquad\square$

**Lemma B.4** (Descent curve on $O_N(\mathbb{R})$). *Let $D_1, D_2 \in \mathbb{R}^{N \times N}$ be fixed diagonal positive definite matrices, each of them with distinct eigenvalues, and define*

$$\phi : O_N(\mathbb{R}) \to \mathbb{R}, \quad \phi(Q) := \|D_1 Q D_2\|^2 = \text{tr}(D_1^2 Q D_2^2 Q^\top).$$

*Then, for almost every $Q_0 \in O_N(\mathbb{R})$ there exists an analytic curve $Q : [0, 1] \to O_N(\mathbb{R})$ such that*

- $Q(0) = Q_0$,

- $Q(1) \in \underset{Q \in O_N(\mathbb{R})}{\arg\min} \ \phi(Q)$,

- $\frac{d}{dt}\phi(Q(t)) \leq 0$.

*Proof.* First, we observe that the structure of all stationary and in particular, of global minima and saddle points is studied in (Brockett, 1989, Theorem 4). From this last result all the minima are global minima, and all saddle points are strict under the hypothesis that all the eigenvalues of both $D_1$ and $D_2$ are distinct. Let us now consider the Riemannian gradient flow

$$\dot{Q} = -\text{grad}\,\phi(Q) = -QD_2^2 Q^\top D_1^2 Q + D_1^2 Q D_2^2,$$

where grad is the gradient induced by the Frobenius inner product on $O_N(\mathbb{R})$. We now use (Bah et al., 2022, Theorem 6.3), which guarantees almost sure (with respect to the initial condition) convergence of the gradient flow to the local minima. Since we know that all local minima are global, we get the desired thesis.

Moreover, the solution curve $Q(t)$ is analytic because the vector field $\text{grad}\,\phi(Q)$ is analytic. $\qquad\square$

Using Lemma B.1, it is possible to prove the main result:

### B.1 Proof of Theorem 4.2

*Proof.* Without loss of generality, we assume $C = 1$ in Definition 4.1. Moreover, we can exclude the case $Z^L \neq Y$, since in that case $\mathscr{L} = +\infty$; therefore we assume $Z^L = Y$. To understand the existence and the structure of the minimizers, we can proceed iteratively. To this end, denote by $\mathcal{S}_l = \mathcal{S}_l(Z^{l-1}, Z^{l+1}) = \{Z^l \text{ that satisfy (I), (II) and (III)}\}$ for fixed, $Z^{l-1}$ and $Z^{l+1}$. Note in particular that

$$\mathcal{S}_l(Z^{l-1}, Z^{l+1}) \subseteq \mathcal{S}_{loc}(\sigma_{l+1}^{-1}(Z^{l+1})), \tag{18}$$

where we refer to equation 15 for the definition of $\mathcal{S}_{loc}(\sigma_{l+1}^{-1}(Z^{l+1}))$. As a consequence we study the infimum in $Z^1$ and observe the following set of inequalities

$$\begin{aligned}
\inf_{Z^1 \in \mathcal{S}_1} \mathscr{L}(Z^L, \dots, Z^1) &\geq \min_{Z^1 \in \mathcal{S}_{loc}(\sigma_2^{-1}(Z^2))} \mathscr{L}(Z^L, \dots, Z^1) \\
&= \frac{1}{2} \sum_{l=2}^{L-1} \|\sigma_{l+1}^{-1}(Z^{l+1})Z^{l,+}\|^2 + \frac{\text{rank}(\sigma_2^{-1}(Z^2))}{2} \\
&\geq \frac{1}{2} \sum_{l=2}^{L-1} \|\sigma_{l+1}^{-1}(Z^{l+1})Z^{l,+}\|^2 + \frac{\text{rank}(Y)}{2},
\end{aligned} \tag{19}$$

where we have used first equation 18, second, Lemma B.1, which proves the existence of the minimum in $\mathcal{S}_{loc}(\sigma_2^{-1}(Z^2))$ and characterizes its value, and third, condition (III) in Definition 4.1, which guarantees the last inequality when applied to $\text{rank}(\sigma_2^{-1}(Z^2))$. Next, we focus on the lower bound in equation 19 and define the following function, depending only on $Z^2, \dots Z^L$:

$$\mathscr{L}_1(Z^L, \dots, Z^2) := \frac{1}{2} \sum_{l=2}^{L-1} \|\sigma_{l+1}^{-1}(Z^{l+1})Z^{l,+}\|^2 + \frac{\text{rank}(Y)}{2}.$$

Thus, we proceed optimizing $\mathscr{L}_1$ with respect to $Z^2$ as we did for $\mathscr{L}$ with respect to $Z^1$ and obtain

$$\begin{aligned}
\inf_{\substack{Z^1 \in \mathcal{S}_1, \\ Z^2 \in \mathcal{S}_2}} \mathscr{L}(Z^L, \dots, Z^2, Z^1) &\geq \inf_{Z^2 \in \mathcal{S}_2} \mathscr{L}_1(Z^L, \dots, Z^2) \\
&\geq \min_{Z^2 \in \mathcal{S}_{loc}(\sigma_3^{-1}(Z^3))} \mathscr{L}_1(Z^L, \dots, Z^2) \\
&= \frac{1}{2} \sum_{l=3}^{L-1} \|\sigma_{l+1}^{-1}(Z^{l+1})Z^{l,+}\|^2 + \frac{\text{rank}(\sigma_3^{-1}(Z^3))}{2} + \frac{\text{rank}(Y)}{2} \\
&\geq \frac{1}{2} \sum_{l=3}^{L-1} \|\sigma_{l+1}^{-1}(Z^{l+1})Z^{l,+}\|^2 + \frac{2\,\text{rank}(Y)}{2}.
\end{aligned} \tag{20}$$

We continue iteratively by defining every time

$$\mathscr{L}_i(Z^L, \dots, Z^i) := \frac{1}{2} \sum_{l=i+1}^{L-1} \|\sigma_{l+1}^{-1}(Z^{l+1})Z^{l,+}\|^2 + \frac{i\,\text{rank}(Y)}{2}.$$

Finally, we obtain

$$\inf_{Z^i \in \mathcal{S}_i \,\forall i=1,\dots,L} \mathscr{L}(Z^L, \dots, Z^2, Z^1) \geq \frac{L-1}{2} \text{rank}(Y).$$

Now we **claim** that there exists some $\bar{Z} \in \mathcal{S}$ such that $\mathscr{L}(\bar{Z}) = \frac{(L-1)}{2} \text{rank}(Y)$. Note that, as a consequence of the claim, we have that the minimum in $\mathcal{S}$ exists and that all the inequalities in equation 19 and equation 20 are equalities. In particular, because of Lemma B.1 any minimum $\bar{Z}$ has to satisfy

$$\bar{Z}^l = O_l \sigma_{l+1}^{-1}(\bar{Z}^{l+1}) \qquad \forall l = 1, \dots, L-1, \tag{21}$$

for some sequence of matrices $O_l \in \mathbb{R}^{n_l \times n_{l+1}}$ such that $O_l^\top O_l = I$. We conclude by observing that $Y = Z^L$ satisfies DNC1 and so, applying equation 21 backwards from $L$ to $1$, also all the other layers $\bar{Z}^l$, at the minima, satisfy DNC1.

We are left to prove the **claim**. To this end, recall that $y_j = e_{i_j}$ for any $j$ and $\sigma_l(0) = 0$ for any $l$. So, $\sigma_L^{-1}(y_j) = \alpha_L e_{i_j}$ for any $j$ where $\alpha_L$ is a constant different from zero and independent of $j$. In particular we can consider $O_{L-1}^T = [\mathrm{Id}, 0]$ and $\bar{Z}^{L-1} = O_{L-1}\sigma_L^{-1}(Y) = \alpha[Y, 0]^T$ and iteratively do the same for any $l$. The obtained $\bar{Z}$ is then included in $\mathcal{S}$ because each layer $Z^l$, as its counterimage $\sigma_l^{-1}(Z^l)$, are a rescaled version of $Y$. In particular, conditions (I),(II),(III) are satisfied. Moreover, by its definition, $\|\sigma_{l+1}^{-1}(\bar{Z}^{l+1})\bar{Z}^{l,+}\|^2 = \mathrm{rank}(\sigma_{l+1}^{-1}(\bar{Z}^{l+1})) = \mathrm{rank}(Y)$, yielding $\mathscr{L}(\bar{Z}) = \frac{(L-1)}{2}\mathrm{rank}(Y)$ and concluding the proof of the claim. $\square$

## B.2 Reachability of DNC configuration for almost every initial condition

A natural question arising from the last section is: from which initial configurations can the global minima be reached? In this section, we will prove not only that the DNC1 minima are global, but also that they are reachable from any interpolating initial condition in a related setting thanks to the result in Theorem 4.4.

We are now ready for the main result about the reachability of rank-collapsed configurations through energy-decreasing paths.

## B.3 Proof of Theorem 4.4

*Proof.* The idea is to construct a concatenation of curves layer by layer, where the existence of a curve for the local problem is given in Lemma B.1. Moreover, we notice immediately that according to Lemma B.1, this can be done for Lebesgue almost every $Z^l(0) \in \mathcal{S}$.

We recall the notation $\mathscr{L}(Z) = \sum_{l=1}^{L-1} \mathscr{L}_l$, where $\mathscr{L}_l := \frac{1}{2}\|\sigma_{l+1}^{-1}(Z^{l+1})Z^{l,+}\|^2$. First observe that, from Lemma B.1, there exists a path $\gamma_1(t)$ moving $Z^1(0)$ to an optimal $Z^{1,*}(Z^2(0)) = OZ^2(0)$, for some $O^\top O = I$. After having optimized the first layer, we can optimize the second one. To this end, we consider a path $\gamma_2^2(t)$ moving $Z^2(0)$ to an optimum $Z^{2,*}(Z^3(0))$, which in turn induces a path $\gamma_2^1(t) \in Z^{1,*}(Z^2(t))$ on the first layer, which keeps it optimal. For this second path $(\gamma_2^1(t), \gamma_2^2(t))$, thanks to Lemma B.5 and the hypothesis that $\mathrm{rank}(\sigma_2^{-1}(Z^2(0))) = N$, we have that for all $t \in [0, 1]$ but a finite set, $\mathrm{rank}(\sigma_2^{-1}(\gamma_2^2(t))) = N$, and therefore the function $\mathscr{L}_2(\gamma_2^2(t)) + \mathrm{rank}(\sigma_2^{-1}(\gamma_2^2(t)))^2$ is continuous up to a finite number of removable discontinuities. In the discontinuities $t_i$, as shown in Lemma B.5, $\mathrm{rank}(\sigma_2^{-1}(\gamma_2^2(t_i))) < N$, the jump is down. In the other points, we have

$$\frac{d}{dt}\mathscr{L}(Z^L(0), \ldots, \gamma_2^2(t), \gamma_2^1(t)) = \frac{d}{dt}\mathscr{L}_2(\gamma_2^2(t)) + \frac{d}{dt}\mathrm{rank}(\sigma_2^{-1}(\gamma_2^2(t)))^2 = \frac{d}{dt}\mathscr{L}_2(\gamma_2^2(t)) \leq 0,$$

where the last is because the rank of $\sigma_2^{-1}(\gamma_2^2(t))$ is constant along the path $\gamma_2^2(t)$.

Inductively, after having optimized the first $l-1$ layers, we can define paths $\gamma_l^{l'}(t)$ for all $l' \leq l$ that move $Z^l(0)$ to an optimum while keeping all the previous layers optimal and satisfying the required decrease condition

$$\frac{d}{dt}\mathscr{L}(Z^L(0), \ldots, \gamma_l^l(t), \ldots, \gamma_l^2(t), \gamma_l^1(t)) = \frac{d}{dt}\mathscr{L}_l(\gamma_l^l(t)) \leq 0.$$

By concatenating all of the paths that we have considered we obtain the required path $\gamma$. Note that for the last path, the rank of all layers stays constant for almost all times, while collapsing to $K$ at $t = 1$, therefore again decreasing the objective. $\square$

**Lemma B.5** (Finitely-many zeros of determinant on analytic matrix curves). *Consider an entrywise analytic function $g$ and a piecewise analytic curve $A : [0,1] \to \mathbb{R}^{N \times N}$ with $\mathrm{rank}\, g(A(0)) = N$. Then there exists at most a finite number of times $\{t_1, \ldots, t_k\}$ in which such $\mathrm{rank}\, g(A(t)) \leq N - 1$. In particular, the function $t \mapsto \mathrm{rank}(g(A(t)))$ has just removable discontinuities, i.e., left and right limits exist and $\lim_{t \to t_i^+} \mathrm{rank}(g(A(t))) = \lim_{t \to t_i^-} \mathrm{rank}(g(A(t))) = N > \mathrm{rank}(g(A(t_i)))$.*

*Proof.* We can prove it without loss of generality for a fully analytic curve $A(t)$, and then the same result holds on all subintervals in which $A$ is analytic (note that there are finitely many of them).

The function $t \mapsto \det(g(A(t)))$ is analytic and not constantly zero since $\det g(A(0)) \neq 0$, and therefore, thanks to Lemma A.3, it has a finite number of zeros in $[0, 1]$. Therefore, there can be just a finite number of points in which $\text{rank } g(A(t)) \leq N - 1$. Now, we will prove that the discontinuities are removable. In particular, let $\varepsilon > 0$. Since $\text{rank}(g(A(t))) = N$ for all $t \in [0, 1] \setminus \{t_1, \ldots, t_k\}$, there exists $\delta > 0$ such that no $t_j \in (t_i - \delta, t_i)$. In $(t_i - \delta, t_i)$, $\text{rank}(g(A(t)) \equiv N$ and therefore we proved that the left limit exists and it is equal to $N$. The same can be done for the right limit. Therefore the concatenation of the paths $\gamma$ is continuous and $\mathscr{L} \circ \gamma$ has a finite number of discontinuities $E := \{t_i\}_i$. $\qquad\square$

As a corollary of Theorem 4.4 we have the following:

### B.4 Proof of Corollary 4.5

*Proof.* Consider the curve $\gamma$ from Theorem 4.4 together with its finite number of discontinuities $E := \{t_i\}_i$ in which

$$\lim_{t \to t_i^-} \mathscr{L}(\gamma(t)) = \lim_{t \to t_i^+} \mathscr{L}(\gamma(t)) > \mathscr{L}(\gamma(t_i)).$$

Consider now a sequence of sets $(\{T_{j,k}\}_{j=0,\ldots,k-1})_k$, where $0 < T_{j+1,k} - T_{j,k} \leq \frac{1}{k}$ for all $j$, $T_{0,k} = 0, T_{k-1,k} = 1$ for all $k$, and such that $T_{j,k} \notin E$ for all $j, k$. Define now the sequence of paths

$$\gamma_k(s) = \gamma\Big(\sum_{j=0}^{k-1} T_{j,k} 1_{[T_{j,k}, T_{j+1,k})}(s)\Big).$$

Then, by construction, we have $\gamma_k : [0, 1] \to S$, $\gamma_k(0) = Z(0), \gamma_k(1)$ global minimizer. Moreover, since $\gamma_k(T_{j,k}) = \gamma(T_{j,k})$ and $T_{j,k} \notin E$ we have that the sequence $(\mathscr{L}(\gamma_k(T_{j,k})))_j = (\mathscr{L}(\gamma(T_{j,k})))_j$ is decreasing. Finally, we show uniform convergence to the $\gamma$ constructed above, by using the fact that $\gamma$ is piecewise analytic and therefore Lipschitz:

$$\sup_{s \in [0,1]} \|\gamma_k(s) - \gamma(s)\| = \max_{j=0,\ldots,k-1} \sup_{s \in [T_{j,k}, T_{j+1,k}]} \|\gamma_k(s) - \gamma(s)\|$$

$$= \max_{j=0,\ldots,k-1} \sup_{s \in [T_{j,k}, T_{j+1,k}]} \|\gamma(T_{j,k}) - \gamma(s)\|$$

$$\leq L \max_{j=0,\ldots,k-1} |T_{j+1,k} - T_{j,k}| \leq \frac{L}{k} \to 0. \qquad (\gamma \text{ Lipschitz})$$

Therefore $\gamma_k$ converges to $\gamma$ uniformly. $\qquad\square$

*Remark* B.6 (Intuition on Corollary 4.5). We highlight that an interpretation of Corollary 4.5, is in terms of numerical schemes. In particular, since $\gamma_k : [0, 1] \to \mathcal{S}$ is piecewise constant, we can think of it as an ordered sequence of $k$ points $\Gamma_k := (\gamma_k(0), \gamma_k(T_{1,k}), \ldots, \gamma_k(T_{k-1,k}), \gamma_k(1))$, where $\gamma_k(0) = Z(0)$ and $\gamma_k(1) \in \arg\min_{Z \in \mathcal{S}} \mathscr{L}$. We know from Corollary 4.5 that $0 < T_{j+1,k} - T_{j,k} := \Delta_{j,k} < \frac{1}{k}$, and we can think of $\Delta_{j,k}$ as the stepsize of an optimization scheme. With this heuristic intuition in mind, the result proven in the corollary essentially states that for Lebesgue-almost every initialization $Z(0) \in \mathcal{S}$, there exists a numerical scheme with **arbitrarily small** maximal stepsize such that $\mathscr{L}$ decreases along it. Notice that, if there was any loss barrier between $Z(0)$ the the global minimizers of $\mathscr{L}$ on $\mathcal{S}$, then this would not be possible: in particular, if there was a barrier between initialization and minimizers, then there would exist a small enough stepsize such that $\mathscr{L}$ would need to increase along the sequence.

### B.5 Stability of the DNC1 Configurations and Proof of Theorem 4.6

*Proof.* We can without loss of generality assume $Z^L = Y$, otherwise the condition is trivially satisfied. Therefore, we can assume $\mathcal{S}$ contains the interpolating condition $Z^L = Y$ as a constraint. Define and consider a sequence $Z_\lambda \to Z^* \in \mathcal{S}$ for $\lambda \to \infty$, $Z_\lambda \in \mathcal{S}$. Since $\mathcal{S}$ contains the interpolating condition, we have

$\mathscr{L}_\lambda(Z^*) < +\infty$. Define $M := \liminf_{\lambda \to +\infty} \mathscr{L}_\lambda(Z_\lambda) = \lim_{m \to +\infty} \mathscr{L}_{\lambda_m}(Z_{\lambda_m})$ for a converging subsequence $\lambda_m$ (we can exclude the case $M = +\infty$ cause the required inequality is trivially satisfied). Then, thanks to the lower semicontinuity of $\mathscr{L}_\lambda$ we get

$$\mathscr{L}(Z^*) \leq \liminf_m \mathscr{L}(Z_{\lambda_m}) = \liminf_{m \to \infty} \frac{1}{2} \sum_{l=1}^{L-1} \|\sigma_{l+1}^{-1}(Z_{\lambda_m}^{l+1})Z_{\lambda_m}^{l,+}\|^2 \leq \liminf_m \mathscr{L}_{\lambda_m}(Z_{\lambda_m}) < +\infty.$$

As a recovery sequence for the $\limsup$ condition we take $Z_\lambda \equiv Z^*$, in fact we have

$$\limsup_{\lambda \to \infty} \mathscr{L}_\lambda(Z^*) = \lim_{\lambda \to \infty} \mathscr{L}_\lambda(Z^*) = \mathscr{L}(Z^*).$$

Moreover, let $\lambda_n \to +\infty$ with $Z_n \in \arg\min_{Z \in \mathcal{S}} \mathscr{L}_{\lambda_n}(Z)$ and that it admits an accumulation point $Z^* \in \mathcal{S}$. Then there exists a convergent subsequence $Z_{n_j} \to Z^*$. For every $(\mathscr{Z}_j)$ recovery sequence for a generic $\mathscr{Z}$ we have, for the definition of $\Gamma$-convergence and the fact that $Z_{n_j}$ is a minimizer of $\mathscr{L}_{\lambda_{n_j}}$,

$$\mathscr{L}(Z^*) \leq \liminf_j \mathscr{L}_{\lambda_{n_j}}(Z_{n_j}) \leq \liminf_j \mathscr{L}_{\lambda_{n_j}}(\mathscr{Z}_j) \leq \limsup_j \mathscr{L}_{\lambda_{n_j}}(\mathscr{Z}_j) \leq \mathscr{L}(\mathscr{Z}),$$

which implies $\mathscr{L}(Z^*) \leq \mathscr{L}(\mathscr{Z})$ for every $\mathscr{Z} \in \mathcal{S}$, therefore $Z^*$ is a minimizer. $\qquad\square$

## C  Global Minima of Representation Cost for Residual Networks

For residual networks, the representation of the layer is given by

$$Z^{l+1} = \sigma_{l+1}(W_{l+1}Z^l) + Z^l$$

and therefore we have

$$W_{l+1} = \sigma_{l+1}^{-1}(Z^{l+1} - Z^l)Z^{l,+},$$

giving the representation cost

$$\mathscr{L}^{res}(Z) := \begin{cases} +\infty, & \text{if } f(\Theta; X) \neq Y \\ \frac{1}{2}\|\sigma_L^{-1}(Z^L)Z^{L-1,+}\|^2 + \frac{1}{2}\sum_{l=2}^{L-2}\|\sigma_{l+1}^{-1}(Z^{l+1} - Z^l)Z^{l,+}\|^2, & \text{otherwise.} \end{cases}$$

The proof of the main result relies on the following lemma:

**Lemma C.1** (Optimal single layer for ResNet representation cost)**.** *Let $\mathscr{L}_{loc}^{res}(B) = \frac{1}{2}\|g(A - B)B^+\|^2$, where $A \in \mathbb{R}^{n \times N}$ is a fixed matrix and $g : \mathbb{R} \to \mathbb{R}$ is entrywise nonlinear with $g(x) = 0 \iff x = 0$. Then, for $n \leq m$, we have that*

$$\min_{B \in \mathbb{R}^{m \times N}} \mathscr{L}_{loc}^{res}(B) \quad \text{s.t.} \quad g(A - B)B^+B = g(A - B)$$

*is attained at the point $B^* = A$.*

*Proof.* Notice that $\mathscr{L}_{loc}^{res}(B) \geq 0$ and $\mathscr{L}_{loc}^{res}(B) = 0$ if and only if $g(A - B)B^+ = 0$. Then we have because of the constraint that

$$0 = g(A - B)B^+B = g(A - B).$$

So the only point attaining the minimum satisfies $g(A - B) = 0$ which implies $B^* = A$. $\qquad\square$

In this setting we have the following result about global minima:

**Theorem C.2** (DNC1 is optimal for constrained representation cost)**.** *Let $\sigma_l$ be a sequence of entrywise nonlinearities for which $\sigma_l(x) = 0 \iff x = 0$ and assume intermediate widths satisfy the condition*

$K \leq n_L = n_{L-1} = \cdots = n_1 = N$. *Then, for almost every $Z^1 = \sigma_1(W_1 X)$, the global minima of the optimization problem*

$$\min_{Z^2,\ldots,Z^L} \mathscr{L}^{res}(Z)$$

$$\text{s.t.} \quad \sigma_{l+1}^{-1}(Z^{l+1} - Z^l)Z^{l,+}Z^l = \sigma_{l+1}^{-1}(Z^{l+1} - Z^l), \quad \forall l = 1,\ldots,L-2$$

$$\sigma_L^{-1}(Z^L)Z^{L-1,+}Z^{L-1} = \sigma_L^{-1}(Z^L)$$

$$s_i(Z^{L-1}) \leq C s_i(\sigma_L^{-1}(Y)), \quad \forall i = 1 \ldots, r_L$$

*satisfy* DNC1 *for all intermediate layers $l = 1,\ldots,L$.*

*Proof.* Let's define with $\mathcal{S}$ the full constraint set and with $\mathcal{S}_l = \mathcal{S}_l(Z^1,\ldots,Z^{l-1},Z^{l+1},\ldots,Z^L)$ the constraint set on $Z^l$ with the other variables fixed. Notice that we can first minimize in $Z^2$ the problem

$$\arg\min_{Z^2 \in \mathcal{S}_2} \mathscr{L}^{res}(Z^L,\ldots,Z^1) = \arg\min_{Z^2 \in \mathcal{S}_2} \|\sigma_3^{-1}(Z^3 - Z^2)Z^{2,+}\|^2$$

$$\text{s.t.} \quad \sigma_3^{-1}(Z^3 - Z^2)Z^{2,+}Z^2 = \sigma_3^{-1}(Z^3 - Z^1)$$

whose minima is given by Lemma C.1 at $Z^{2,*}(Z^3) = Z^3$ and minimal value equal to zero. Therefore, we can now minimize in $Z^3$ the representation cost

$$\min_{Z^3,\ldots,Z^L} \mathscr{L}^{res}(Z^L,\ldots,Z^2,Z^{2,*}(Z^3)) = \frac{1}{2}\|\sigma_L^{-1}(Z^L)Z^{L-1,+}\|^2 + \frac{1}{2}\sum_{l=3}^{L-1}\|\sigma_{l+1}^{-1}(Z^{l+1} - Z^l)Z^{l,+}\|^2$$

$$\text{s.t.} \quad \sigma_{l+1}^{-1}(Z^{l+1} - Z^l)Z^{l,+}Z^l = \sigma_{l+1}^{-1}(Z^{l+1} - Z^l) \quad \forall l = 3,\ldots,L-1$$

$$\sigma_L^{-1}(Z^L)Z^{L-1,+}Z^{L-1} = \sigma_L^{-1}(Z^L)$$

$$s_i(Z^{L-1}) \leq C s_i(\sigma_L^{-1}(Y)), \quad \forall i = 1 \ldots, K.$$

Notice that the situation now is the same as the initial one, but with one layer less. Therefore, we can do the same thing for $Z^3$ using Lemma C.1 (with $g = \sigma_l^{-1}$) to obtain $Z^{3,*}(Z^4) = Z^4$. By iterating this procedure for all layers, we arrive at

$$\min_{Z^{L-1},Z^L} \frac{1}{2}\|\sigma_L^{-1}(Z^L)Z^{L-1,+}\|^2$$

$$\text{s.t.} \quad Z^L = Y$$

$$\sigma_L^{-1}(Z^L)Z^{L-1,+}Z^{L-1} = \sigma_L^{-1}(Z^L)$$

$$s_i(Z^{L-1}) \leq C s_i(\sigma_L^{-1}(Y)), \quad \forall i = 1 \ldots, K.$$

By using Lemma B.1 we get that

$$Z^{L-1,*} = O_l \sigma_L^{-1}(Y), \quad O_l^\top O_l = I.$$

Therefore, $Z^{L-1,*}$ satisfies DNC1 cause $\tilde{Y} = \sigma_L(Y)$ satisfies DNC1. Recursively backward, we have by Lemma C.1 that

$$Z^{L-2,*} = Z^{L-1,*},$$

which is therefore also collapsed. In the same way, by backsubstituting the optimal point, we get

$$Z^{L-1,*} \quad \text{collapsed}$$

$$Z^{L-2,*} = Z^{L-1,*}$$

$$\vdots$$

$$Z^{1,*} = Z^{2,*}$$

and therefore

$$Z^{L-1,*} = Z^{L-2,*} = \cdots = Z^{1,*}$$

and so they all satisfy DNC1. $\qquad\square$

*Remark* C.3. (Stability of minima for ResNets) We highlight that the same $\Gamma$-convergence result as the one presented in Theorem 4.6 can be stated also for ResNets, and the proof is exactly the same as the one presented in Appendix B.5. More precisely, given $\phi : \mathcal{S} \to \mathbb{R}_+$ lower semicontinuous, we define

$$\mathscr{L}_\lambda^{res}(Z) = \begin{cases} \frac{1}{2}\|\sigma_L^{-1}(Z^L)Z^{L-1,+}\|^2 + \frac{1}{2}\sum_{l=1}^{L-2}\|\sigma_{l+1}^{-1}(Z^{l+1}-Z^l)Z^{l,+}\|^2 + \frac{\phi(Z)}{\lambda}, & \text{if } Z^L = Y, \\ +\infty, & \text{otherwise.} \end{cases}$$

Then for $\lambda \to +\infty$, $\mathscr{L}_\lambda^{res} \to \mathscr{L}^{res}$ in the $\Gamma$-convergence sense, analogously to Theorem 4.6. This will also be showcased numerically in Appendix D.

# D   Additional Numerical Experiments

In this section, we include additional numerical results. All experiments were performed on a Single NVIDIA A100 (80GB) GPU.

## D.1   Testing the set of constraints Definition 4.1

In this section, we present a numerical experiment that demonstrates the reasonableness of the constraints in Definition 4.1 in standard training settings. In particular, we trained a feedforward network in the same setting used in Section 5.2. The network has been trained beyond interpolation, and we tracked the ranks $\text{rank}\,\sigma_l^{-1}(Z^l)$ and singular value ratios $\frac{s_i(Z^l(t))}{s_i(\sigma_{l+1}^{-1}(Z^{l+1}(t)))}$ for all layer $l$ and indices $i$ over training. We recall that the second constraint in Definition 4.1 requires that $\frac{s_i(Z^l}{s_i(\sigma_{l+1}^{-1}(Z^{l+1})} \leq C_{l+1}$, and the third one requires that $\text{rank}(\sigma_l^{-1}(Z^l)) \geq \text{rank}(Y)$ (which is $\text{rank}(Y) = 10$ in this example).

In Figure 4 we present numerical results: on the left figure we can observe that the ratios present in the second constraint of Definition 4.1 are indeed bounded for each singular value index and layer, uniformly across epochs. On the right figure, we can similarly observe that the third constraint is satisfied, i.e., that $\text{rank}(\sigma_l^{-1}(Z^l)) \geq \text{rank}(Y) = 10$ for all times.

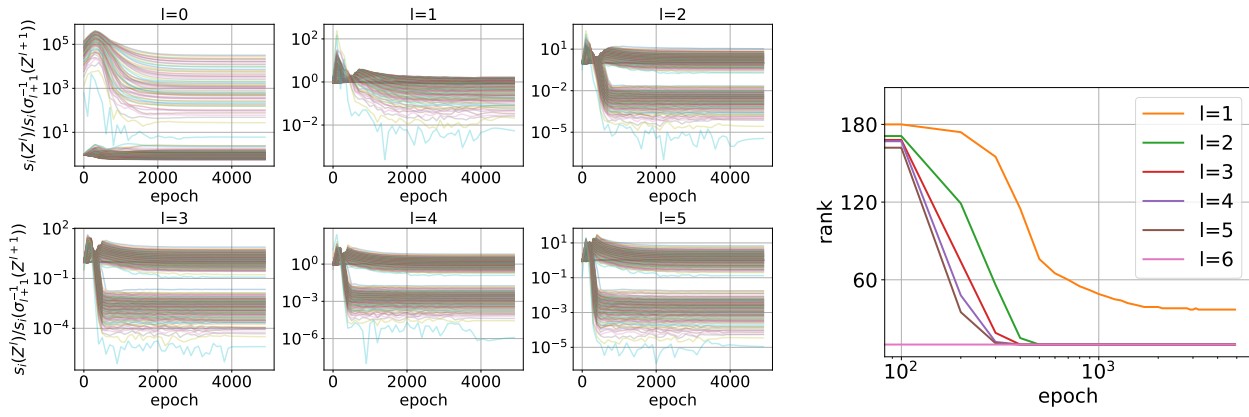

Figure 4: Numerical experiment evidencing the last two constraints in Definition 4.1. On the left figure, we plot the ratio $\frac{s_i(Z^l)}{s_i(\sigma_{l+1}^{-1}(Z^{l+1}))}$ across epochs and for every layer. On the right figure, we plot $\text{rank}(\sigma_l^{-1}(Z^l))$ across iterations and for all layers.

## D.2   Residual Networks

In this section we will show numerical results concerning Theorem C.2. The setting is the same of Section 5, with $L = 6$ layers (first and last feedforward, and the 4 in the middle are residual). From Figure 5, we can

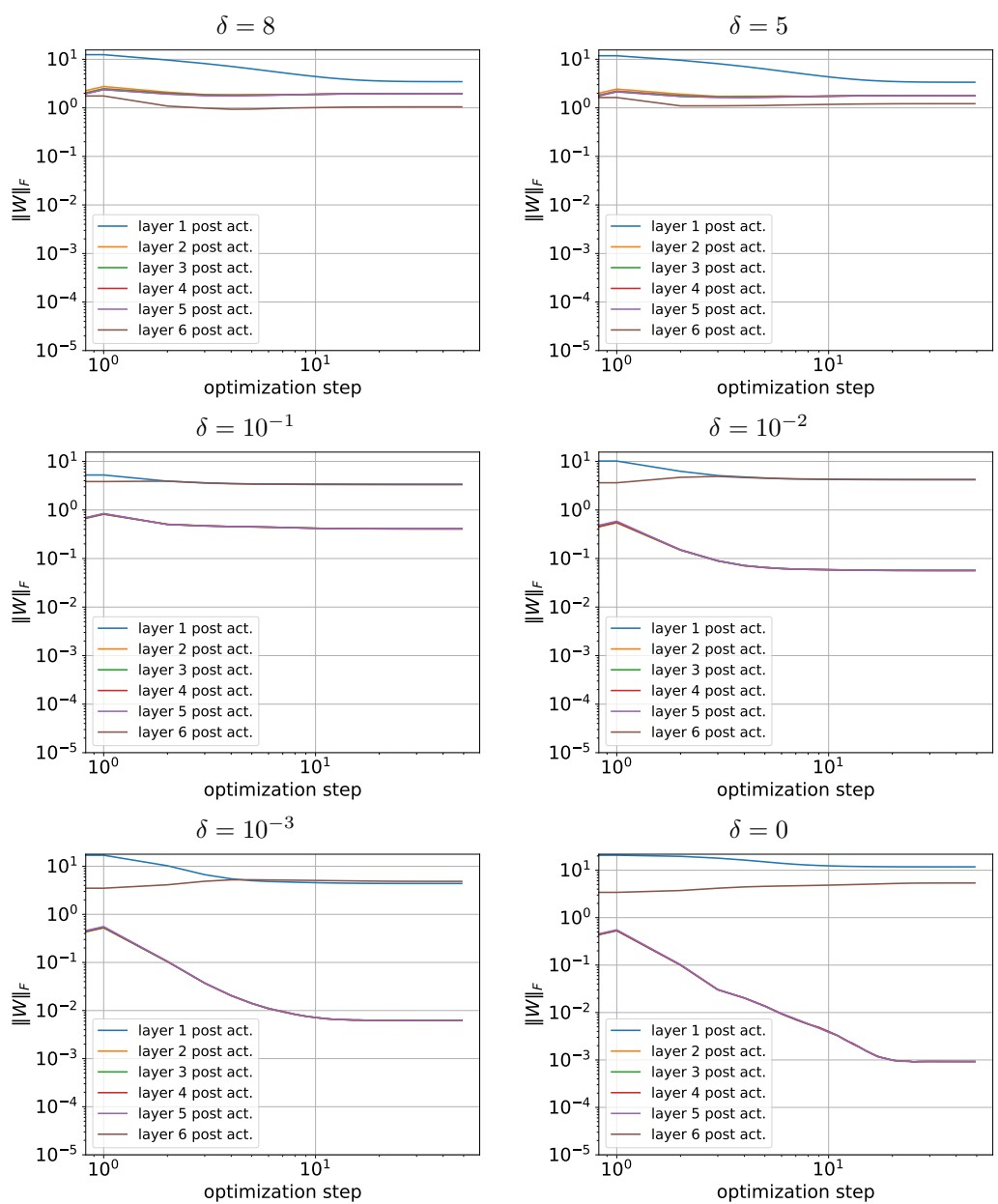

Figure 5: Norm of the intermediate weights matrices for a for $L = 6$ layer ResNet during training. As discussed in Remark C.3, we observe convergence to the structured minima discussed in Theorem C.2 for $\delta = \frac{1}{\lambda}$ (as in Equation (13)).

observe the evolution of the norm of the weights during optimization, also for varying $\delta = \frac{1}{\lambda}$ in Equation (13). As we can see, for $\delta$ going to zero, all the weight matrices converge to zero as expected from Theorem C.2, but the first and the last ones, which are feedforward layers.

## D.3    Visual Representation of Neural Collapse

In this section we will include the evolution of the gram matrices $G^l = Z^{l,\top} Z^l$ for all layers at different phases of training. Notice that DNC1 happens if and only if $G^l$ has block structure. In fact, we have the following characterization:

**Proposition D.1** (Characterization of DNC1 features through covariance)**.** *The following are equivalent:*

1. *$Z \in \mathbb{R}^{n \times N}$ satisfied NC1;*

2. *$G = Z^\top Z =$*
$$\left[ \begin{array}{c|c|c|c|c} \alpha_{11} 1_{N_1} 1_{N_1}^\top & \alpha_{12} 1_{N_1} 1_{N_2}^\top & \alpha_{13} 1_{N_1} 1_{N_3}^\top & \dots & \alpha_{1K} 1_{N_1} 1_{N_K}^\top \\ \hline \alpha_{21} 1_{N_2} 1_{N_1}^\top & \alpha_{22} 1_{N_2} 1_{N_2}^\top & \alpha_{23} 11^\top & \dots & \alpha_{2K} 11^\top \\ \hline \vdots & \dots & \ddots & \dots & \alpha_{K-1,K} 11^\top \\ \hline \alpha_{K1} 1_{N_K} 1_{N_1}^\top & \alpha_{K,2} 11^\top & \alpha_{K,3} 11^\top & \dots & \alpha_{K,K} 11^\top \end{array} \right]$$ *has block structure.*

*Proof.* For simplicity, throughout the proof, we will denote with $z_i^k$ the $i$-th vector of the $k$-th class. Assume first that $Z$ satisfies NC1. Then we have that $z_i^k = z^k$ for all $i = 1, \dots, N_k$ and

$$Z^\top Z = \begin{bmatrix} 1_{N_1} z^{1,\top} \\ 1_{N_2} z^{2,\top} \\ \vdots \\ 1_{N_K} z^{K,\top} \end{bmatrix} \begin{bmatrix} z^1 1_{N_1}^\top, z^2 1_{N_2}^\top, \dots z^K 1_{N_K}^\top \end{bmatrix} = \left( \langle z^i, z^j \rangle 1_{N_i} 1_{N_j}^\top \right)_{i,j=1,\dots,K},$$

which has the required structure of $Z^\top Z$ with $\alpha_{ij} = \langle z^i, z^j \rangle$. Vice versa, assuming that $Z^\top Z$ has the required structure, we have

$$\alpha_{kk} = \langle z_i^k, z_j^k \rangle, \quad \forall i, j = 1, \dots, N_k \text{ and } \forall k = 1, \dots, K.$$

This implies that

$$0 = \langle z_i^k, z_j^k - z_m^k \rangle, \quad \forall i, j, m = 1, \dots, N_k.$$

This, in turn, implies by definition of orthogonal projection that

$$z_i^k = P_{\mathbb{A}^k}^\perp(0), \quad \forall i = 1, \dots, N_k \text{ and } \forall k = 1, \dots, K,$$

where $\mathbb{A}_k = \mathrm{Aff}(\{z_j^k\}_j)$ is the affine subspace passing through the points of class $k$. By uniqueness of the orthogonal projection we get

$$z_i^k = z_j^k = P_{\mathbb{A}^k}^\perp(0) \quad \forall i, j = 1, \dots, N_k \text{ and } \forall k = 1, \dots, K,$$

which means $Z$ satisfies NC1. $\qquad\square$

Given this premise, in the following experiments we will plot the Gram matrices $G_l = Z^{l,\top} Z^l$ for any $l = 0, \dots, L-1$ in the case of a $L = 15$ layer neural network trained in the setting of Section 5. The emergence of the block structure is depicted in Figure 6.

Moreover, we sampled a random plane in each feature space, i.e., for any $l$ we sample a random orthogonal matrix $B^l \in \mathbb{R}^{2 \times n_l}, B^l B^{l,\top} = I_2$. Figure 7 shows the projection of the feature matrices $Z^l$ on the plane spanned by the rows of $B^l$, i.e. $B^l Z^l$, for which we plot all the columns as points in $\mathbb{R}^2$, allowing us to pictorially see the deep-neural (rank) collapse in all intermediate layers in this random subspace.

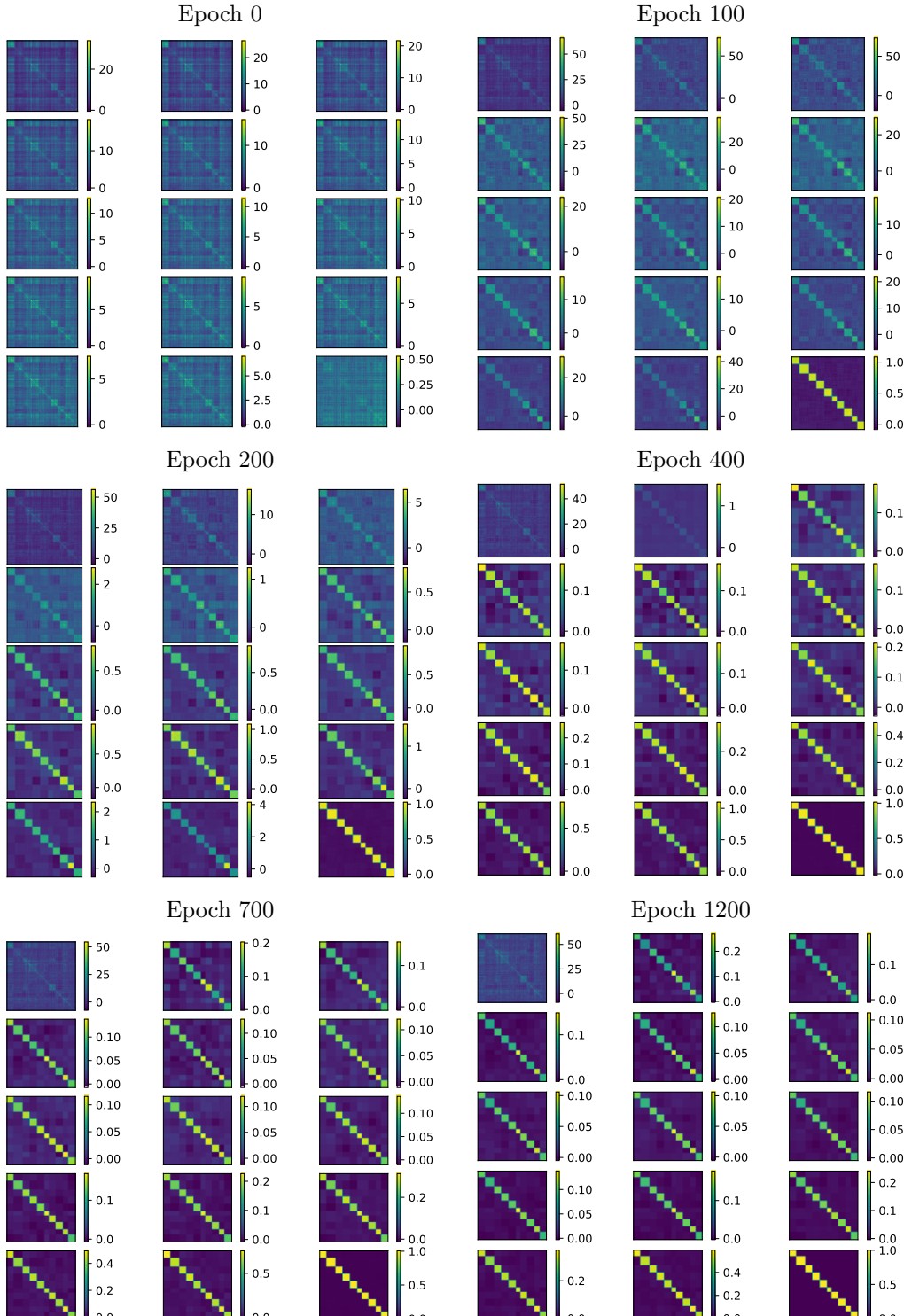

Figure 6: Emergence of the block structure in $Z^{l,\top}Z^l$ characterizing DNC1 during training. Layers for each epoch are ordered from left to right and from top to down.

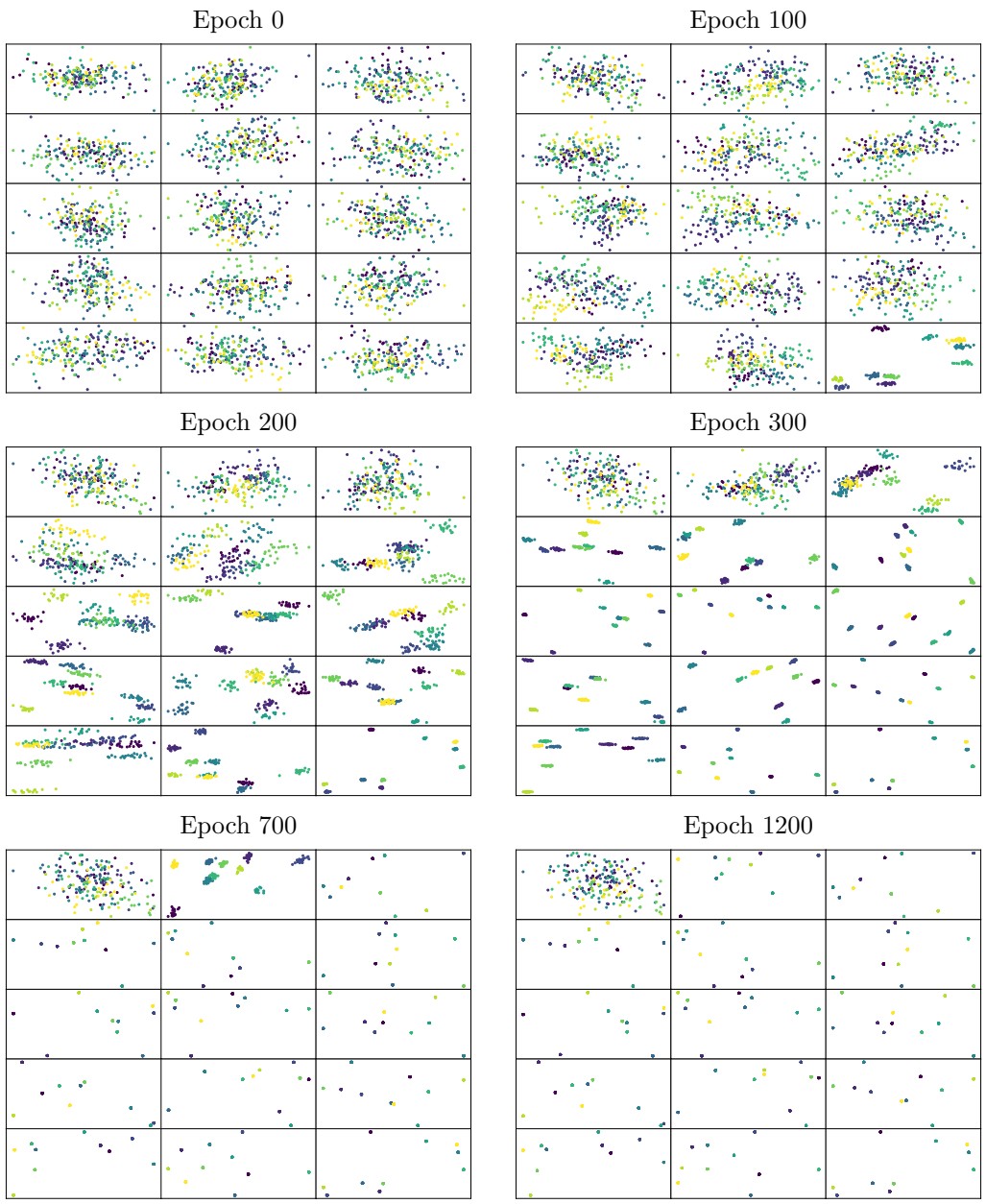

Figure 7: Neural collapse of features in a random $d = 2$ dimensional subspace, where each color represents a class of MNIST. Layers for each epoch are ordered from left to right and from top to down.

