# OpenReview forum: "Provable Emergence of Deep Neural Collapse and Low-Rank Bias in $L^2$-Regularized Nonlinear Networks"
_TMLR — Accepted by TMLR_

### Review · Reviewer_QjdM · 2026-04-10

**Summary Of Contributions:**

The paper studies the connection between Neural Collapse (and its variant called deep Neural Collapse -- DNC -- for intermediate layers) and the emergence of low-rank bias. Specifically, the authors theoretically establish a link between the emergence of DNC1 and its implications for the evolution of weights ranks at critical points. Additionally, the analysis establishes the optimality of DNC1 in a ocnstrained representation cost setting and show that for almost every interpolation initialization there exists a cont path without loss barriers to global optimal satisfying DNC1. The theoretical results are backed with the empirical validation on a synthetic datasets to validate the claims made by the authors.

**Additional Comments:**

Since the whole work is focused on the training of NNs with weight-decay I would like to understand the author's position on the low-rankness in terms of training without weight decay? Would you expect these training to ever exhibit this kind of structure at all? It's by no means a weakness of this work to focus on WD only, but a quick discussion about this would be interesting to read given your expertise in this domain.

As the proofs of low-rankness are tied to DNC-1 which itself is tied to Terminal Phase Training (TPT) is my understanding correct, that reaching TPT/interpolation is a necessary condition to observe low-rankness given the theoretical results that you present?

**Audience:**

Yes

**Audience Explanation:**

The topic of representation learning and its influence on the structure of learned weights and representation is an important topic. The paper helps understand the connection between the empirically observed Deep Neural Collapse and the low-rank bias of model's weights which gives a better understanding of the learning dynamics.

**Claims And Evidence:**

Yes

**Claims Explanation:**

All the claims are supported with clear theoretical and empirical results. The authors have paid extra attention to precisely define the scope of their arguments stressing the assumptions and constraints necessary for the results to hold.

**Requested Changes:**

I'd like to suggest two minor changes to the authors:
1) The current structure of the work misses the related section which makes it a bit hard to position the work correctly within the field. While I do acknowledge that the authors have already tried to position their work within the field (mostly in the Introduction section) I believe that a proper section would improve the overall readability of the work and would help understand the contributions of this work better.

2) Since the work is mostly focused on the low-rank bias and its connection to DNC-1 I believe that introducing a clear definition of these two concepts would help the reader to navigate the work. Especially, the concept of low-rank is quite unclear to me in my current understanding the rank of the final solution directly depends on the number of clusters (or classes) marked as $K$ in the work. Since in the most setups $K << d$, where $d$ is the dimension of weights matrix (suppose square MLPs for simplicity) than I intuitively understand what this low-rankness mean but what would happen when $K \approx d$ or $K > d$? I mean the analysis still holds, but saying about low-rankness in this case seems a bit artificial to me. Thus, I believe that clearly defining the "low-rankness" upfront would help a lot.

---

> ### Author Response · Authors · 2026-05-08
>
> We first of all thank the reviewer for their comments. We respond to the different points in order, sometimes referring to the revised version of the manuscript.
>
> **Requested changes**
> 1. Following the reviewer's suggestion, we added a "related work" section and moved our discussion there to improve readability.
> 2. We thank the reviewer for the comment. We remark that a precise definition of DNC1 is already included in Definition 3.3, while we agree on the proposed intuition on the definition of low-rank. In particular, our results are meaningful when $K \ll d$. To improve the work readability, we include a definition of what is meant by "low-rank" in light of the results in section 4. In particular, we say a family of matrices $X_d \in \mathbb R^{d \times d}$ is low-rank if $\mathrm{rank}(X_d) = o(d)$. In light of this definition, our results imply that the global minimizers of Theorem 4.2 have rank $O(1)$, as they are constant with respect to the intermediate widths of the network. We, however, remark that the setting in which we pose ourselves in section 4, given the assumption of piramidality of the architecture, is the one in which $K \leq d$. We have discussed this in the revised version (Definition 3.7 and the following lines).
>
> **Additional comments**
> Thanks for this point. Our current analysis relies strongly on the presence of weight decay in the model and does not provide any insight into the emergence of low-rank structures in DNNs trained without it. In particular, the properties we derive are properties of the landscape itself, not linked to the particular dynamics chosen for the optimization process.
> In practice, we characterize the structure of stationary points of the loss landscape in the limit of infinitesimal regularization. What arises from our analysis is that global minimizers of this limit landscape, under certain restrictions, are in fact all low-rank ($O(1)$ in terms of width), and the intermediate representations satisfy DNC1.
> In our framework, getting rid of regularization destroys this property.
>
> This said, something can also be recovered in the unregularized case.
> Defining $\chi_\lambda:= \arg\min \mathcal L_\lambda$ as the set of global minimizers of the problem in Eq.3 and with $ \chi^* $ the set of global minimizers of the problem in Eq.9 presented in Theorem 4.2, it is easy to notice that $\chi^* \subseteq \chi_0$ (because all minimizers of the limit problem are interpolatory, therefore globall minimizers of the unregularized loss). This inclusion is, however, **strict** in general. For this reason, there are minimizers with the properties described in our manuscript inside the minimizers of the unregularized problem, but the rest is in general made of full-rank matrices.
> In this sense, convergence to points in $\chi^*$ in the unregularized setting could be driven only by properties of the optimization dynamics. More precisely, once interpolation is reached, a regularized landscape would naturally drive the network towards these collapsed representations (in the sense of Theorem 4.4). Differently for the unregularized loss, once interpolatory minimizers are reached, a general gradient-based dynamics would stop regardless of the kind of representation. For this reason, according to our analysis, convergence to low-rank stationary points in the unregularized setting is possible but not necessary; however, we do not rule out that some optimization dynamics may favor this kind of stationary points.
>
> If the reviewer considers it useful, we would be happy to include a remark about this in the updated version.
>
> Regarding the comment on the TPT: yes, in our theoretical analysis, interpolation is necessary for all results in section 4, which are in turn sufficient for low-rankness. In general, however, it could be possible to observe low-rank weights even outside the interpolatory regime.

---

### Review · Reviewer_4EvC · 2026-04-18

**Summary Of Contributions:**

This paper argues that deep neural collapse (DNC1) and low-rank weight matrices are closely related, both driven by $L_2$​ weight decay. The core idea is: when weight decay is small, the cheapest way for a network to fit the data is to collapse same-class samples to a single point at every intermediate layer, and this collapse forces the weight matrices to be low-rank.

**Audience:**

Yes

**Audience Explanation:**

Yes. Neural collapse and implicit low-rank bias are both active areas in deep learning theory, and this paper connect them rigorously for general nonlinear networks rather than simplified UFM-style models. Researchers working on neural collapse (intermediate layers) and representation learning will likely find the results relevant.

**Broader Impact Concerns:**

None. This is a theoretical paper. No Broader Impact Statement is needed.

**Claims And Evidence:**

Yes

**Claims Explanation:**

Yes, largely. The proofs look correct and each experiment targets a specific prediction: Experiment 5.1 confirms Theorem 3.5's qualitative trend that weights approach rank-$K$ as $\sigma$ shrinks and $\lambda$ grows; Experiment 5.2 (MNIST, Figure 2) shows the singular values of trained features match Theorem 4.2's closed-form as $\delta \to 0$; Experiment 5.3 (deeper network, Figure 3) confirms TCV goes to zero and numerical rank drops to $K$, consistent with Theorem 4.4.

**Requested Changes:**

**1. Discuss the scaling of the constant in Theorem 3.5.** The bound involves a term $M_{l,j}$ whose effect is never analyzed. How does $M_{l,j}$ grow for real networks? The experiment in Section 5.1 is run on a controlled Gaussian mixture rather than a real dataset, which sidesteps the contribution of $M_{l,j}$. The authors should either argue theoretically that $M_{l,j}$ stays bounded (e.g., for smooth activations + MSE) or measure it empirically on a real setup such as MNIST.

**2. Empirically verify the assumptions in Definition 4.1.** The experiments confirm the conclusions of Theorem 4.2 but never check whether trained networks satisfy constraints (II) and (III). The authors themselves note these are introduced for the proof, so it is worth checking they actually hold in practice. Concretely, I suggest plotting (a) the singular-value ratios in (II) across layers and epochs, and (b) the rank in (III) during training. This would support the "natural architectural constraints" framing and prove the network satisfying them on its own.

---

> ### Author Response · Authors · 2026-05-08
>
> We first of all thank the reviewer for their comments. We respond to the comments in order, sometimes referring to the revised version of the manuscript.
>
>
> **Requested changes**
>
> 1. We thank the reviewer for pointing this out. We believe it could be more resolutive to go with a theoretical argument, therefore we added a remark in the revised version (Remark 3.8), clarifying that in fact $M_{l,j}$ stays bounded for smooth activation functions and MSE loss on global minimizers. We report here a sketch of the main idea, while we refer to the updated manuscript for further details.
>  Let $ \Theta_\lambda^* $ be a global minimizer of $ \mathcal L_\lambda $, then we have
> $$ \inf_{\Theta} \mathcal L_0(\Theta) + \lambda|| \Theta_\lambda^* ||^2 \leq \mathcal L_\lambda( \Theta_\lambda^* )  \leq \mathcal L_\lambda(\Theta_0^*)= \inf_{\Theta} \mathcal L_0(\Theta) +  \lambda || \Theta_0^\ast ||^2,$$
>
> which indeed implies $ ||\Theta_\lambda^* || \leq ||\Theta_0^* || $ for any $ \Theta_0^* $ global minimizer of $\mathcal L_0$. In particular, defined $\Omega:= \\{ \Theta  : ||\Theta || \leq ||\Theta_0^* || \\}$, if all activation functions and $\ell$ are smooth, then by compactness of $\Omega$ we get boundedness of $H^j$ (there exists a finite radius ball $\mathcal K$ such that $H^j \subseteq \mathcal K$) and therefore by Weierstrass theorem
> $$M_{l,j}(\Theta^*) \leq \sup_{\Theta \in \Omega} M_{l,j}(\Theta)= \sup_{\Theta \in \Omega} \sup_{(\alpha,\beta)\in \mathcal K}\|\nabla_{(z_j,y)}^2\nabla_{W_l} \ell(f_j(\Theta,z^j),\alpha,\beta)\|<+ \infty,$$
> which means in fact that $M_{l,j}$ is in fact uniformly bounded on the set of global minimizers (which is all contained in $\Omega$) and in any compact set of regularization parameters $\lambda \in [0,\lambda_{max}]$.
>
> 2. Following the reviewer's suggestion, we included an experiment in Appendix D.1 of the revised manuscript to numerically verify the reasonableness of constraints $(II)$ and $(III)$, together with a pointer in the main corpus (just before Theorem 4.2).
> As a brief synthesis, we trained a feedforward network beyond the interpolation regime in the setting of section 5.2 (on MNIST). We plot the singular value ratios in $(II)$ for all layers and across iterations, together with the rank in $(III)$. From the numerical test, we can indeed observe that the ratio is uniformly bounded across iterations and indices for each layer, and that the rank in $(III)$ is, in fact, lower bounded by the rank of the target matrix across all iterations. For further details, we refer to the updated manuscript.

---

### Review · Reviewer_jjiB · 2026-05-04

**Summary Of Contributions:**

This article shows that the first deep neural collapse property (DNC1), minimal within-cluster variance of intermediate hidden representations, implies a low-rank bias in the layer weights. The authors establish 3 theoretical results,
1. **Theorem 3.5**: layer weights of a stationary point of the regularized optimization objective (3) are nearby low-rank matrices if the total cluster variation (Def 3.2) in **previous layers** is minimal. This result suggests that the rank-complexity of subsequent layers during training are impacted by variance collapse in preceeding layers.
2. **Theorem 4.2**: The authors construct a representation cost of features (based on Jacot et. al. 2022) and show the global optimality of the DNC1 condition under certain regularity constraints in the feature space.
3. **Theorem 4.4.**: A benign-landscape trajectory with non-increasing loss exists between any interpolation initialization and a global minimum satisfying DNC1. This interesting observation supports the empirical evidence of consistent deep neural collapse on training past the interpolation regime.
Prior theoretical investigations largely tackle the neural collapse phenomena in the last layer and often through the unconstrained-feature model (Mixon et. al. 2020). Rangamani et. al. (2023b)'s empirical evidence suggests that intermediate layers also exhibit feature collapse. This article generalizes the results of Sukenik et. al. (2024a,b) to continuously differentiable non-linear neural architectures.

# Strengths
1. Theorem 3.5 appears to improve on Lemma 3.1 in Galanti et. al. (2023) where rank of the weight gradient is controlled by the batch-size and thus vacuous for larger batch sizes. The motivation of the stationarity condition as a non-linear eigenvalue operator was a convenient mental picture for why low-rank gradients result in low-rank weights.
2. The benign-landscape result (Theorem 4.4) appears novel and provides the strongest theoretical evidence for the ubiquity of intermediate neural collapse observed in Rangamani et. al. (2023b)
3. The theoretical proofs are well presented. To the best of my knowledge the calculations in the appendix are correct.

# Weakness
1. Theorem 4.2 requires two regularity constraints on the features (Def 4.1). Constraints (II) and (III) in Def 4.1 appear specialized to the proof arguments rather than general problem structure. (II) controls singular-value growth across layers while (III) rules out lower-rank optima. While I understand the difficulties of synthesizing theoretical results, the empirical observations of deep neural collapse requires no training constraints. Further, the authors comment "Without considering additional constraints, the minima of $\mathcal{L}$ may not exist" but the discussion of the degeneracy in the absence of constraints is not sufficiently discussed.
2. The assumptions of Theorem 4.2 require invertible layer-wise activation functions. The manuscript as such does not explicitly note this requirement in the abstract or the introduction. I suggest the authors add a clear statement on this in their contributions.

**Additional Comments:**

# Minor Typos and Other Comments
### Editorial: As a reviewer I would have found it helpful if all equations are numbered (as opposed to the current selective numbering).
1. Equations (1) and (2) define bias as trainable parameters, but I think the entirety of the article appears to assume there are no bias vectors in the neural architecture.
2. Page 4 top sentence duplicate wording: “In addition, equation equation 4 immediately implies that..
3. Proof of Proposition 3.1, the calculations are overall correct, few minor remarks -
 - The chain of equalities is not in the sequential order of applying chain rule for gradient computation.
  - I think the authors mean - $\nabla_f \text{loss} z^{l-1}(x)^T$ instead of $\nabla_f \text{loss} z^{l}(x)^T$
 - The authors are switching between matrix derivatives and vector-derivatives back and forth, given the simplicity of the actual argument here, it would be easier to read if the authors picked only one abstraction.
4. Is the subscript index in the summation in equation 9, $l=1$ instead of $l=2$? Equation 12 says $l=1$, text indicates $l=1$ but equation 9 says $l=2$.
5. Equation 10 is a little confusing. It only holds bi-directionally if the features matrix $Z^l$ have full row-rank right? I think the message here is that if one were to first fix the features, then there is a corresponding minimum norm choice of layer weights, given by the right hand side of equation 10, that satisfies the constraint in the left hand side of equation 10. I’d urge the authors to be clearer on this (or correct me if my understanding is wrong).
6. Why can’t equation 12 have $l=1$? With $\| \sigma_1^{-1}(Z^1) X^{+}\|_2$ , is the issue assuming pseudo-invertability of the data matrix? Or simply that the layer-by-layer peeling will encounter an additional constraint at the first layer and thus be sub-optimal?
7. On Page 22, I think the optimal value of the localized objective is $\frac{\text{rank}(A)}{2C}$ rather than $\frac{\text{rank}^2(A)}{2C}$ (check the line “and the optimal value is given by ..” )
8. On Page 24, I think the final lower bound in the line “Finally, we obtain …” should be $\frac{L-1}{2} \text{rank}(Y)$ given the summation from $l=2$ to $L-1$. Admittedly it doesn’t really change the conclusion.

**Audience:**

Yes

**Audience Explanation:**

Understanding implicit bias in neural networks and the impact of neural collapse is of enduring importance in theoretical machine learning.

**Broader Impact Concerns:**

I foresee no broader impact concerns.

**Claims And Evidence:**

Yes

**Claims Explanation:**

The theoretical arguments are largely complete and further our rigorous understanding of the deep neural collapse.

**Requested Changes:**

1. The bound in Theorem 3.5 is informative when the total cluster variation $TCV_{K,l}$ is minimal for $K \ll \min\{n_l, n_{l-1}\}$ the maximum possible rank of the layer weight. The paper would benefit from commenting explicitly on this as the present comparison to sample size $N$ is insufficient for meaningful complexity control. I would also the like the authors to comment on the size of the $M_{l,j}(\theta^\star)$ quantity. For what ground-truth optimal $\theta^*$ does one expect control on $M\_{l,j}(\theta^\star)$.
2. Definition 3.3 identifies DNC1 as the combination of $TCV_{c,l}=0$ *and* the minimizing partition in equation (7) coinciding with the class partition provided by the labels. The proof of Theorem 4.2 establishes that the global minimizer of the constrained objective satisfies $\bar{Z}^l \propto O_l \sigma_{l+1}^{-1}(\bar{Z}^{l+1})$ recursively from $\bar{Z}^l=Y$ for any fixed $\bar{Z}^1$. I think this does imply a class-partition alignment but the proof arguments appear to not verify this explicitly. I would like the authors to bridge this gap (or explain why the proof already meets the required threshold).

---

> ### Author Response · Authors · 2026-05-08
> **Official Comment by Authors 1/2**
>
> We first of all thank the reviewer for their careful comments. We respond to the points in order, sometimes referring to the revised version of the manuscript.
>
> **Weaknesses**
> 1. As suggested by the reviewer, we included in the manuscript a comment just before subsection 4.1, clarifying the sentence *"Without considering additional constraints, the minima $\mathcal L$ of may not exist"*. Moreover, as also requested by reviewer 4EvC, we included a complementary numerical experiment to validate constraints (II) and (III) during training. We include the additional results in the appendix of the revised manuscript (Appendix D.1).
> 2. That's a fair point, thank you for pointing it out. We made this assumption clearer in the abstract and in the introduction of the revised version.
>
> **Requested changes**
> 1. We explicitly included a comment in the revised version clarifying the meaning of low-rank in contexts in which the intermediate dimension is of comparable size with $K$ (Definition 3.6 and the following discussion). As concerns the control of $M_{l,j}$, we added a remark in the revised version as asked also by Reviewer 4EvC (Remark 3.8). In particular, $M_{l,j}$ stays bounded for smooth activation functions and MSE loss on global minimizers (which is the setting of section 4). We report here a sketch of the main idea, while we refer to the updated manuscript for further details.
>  Let $ \Theta_\lambda^\ast $ be a global minimizer of $ \mathcal L_\lambda $, then we have
> $$ \inf_{\Theta} \mathcal L_0(\Theta) + \lambda ||\Theta_\lambda^* ||^2 \leq \mathcal L_\lambda(\Theta_\lambda^\ast) \leq  \mathcal L_\lambda( \Theta_0^* ) =\inf_{\Theta} \mathcal L_0(\Theta) +  \lambda || \Theta_0^\ast ||^2 , $$
> which indeed implies $||\Theta_\lambda^* || \leq ||\Theta_0^* ||$ for any $ \Theta_0^* $ global minimizer of $\mathcal L_0$. In particular, defined $\Omega:= \\{\Theta : ||\Theta || \leq ||\Theta_0^* || \\}$, if all activation functions and $\ell$ are smooth, then by compactness of $\Omega$ we get boundedness of $H^j$ (there exists a finite radius ball $\mathcal K$ such that $H^j \subseteq \mathcal K$) and therefore by Weierstrass theorem
> $$M_{l,j}(\Theta^*) \leq \sup_{\Theta \in \Omega} M_{l,j}(\Theta)= \sup_{\Theta \in \Omega} \sup_{(\alpha,\beta)\in \mathcal K}\|\nabla_{(z_j,y)}^2\nabla_{W_l} \ell(f_j(\Theta,z^j),\alpha,\beta)\|<+ \infty,$$
> which means in fact that $M_{l,j}$ is in fact uniformly bounded on the set of global minimizers (which is all contained in $\Omega$) and in any compact set of regularization parameters $\lambda \in [0,\lambda_{max}]$.
> 2. The result of Theorem 4.2 $\bar Z^l = O_l \sigma_{l+1}^{-1}(\bar Z^{l+1})$ already implies a class partition alignment. This happens because $Y$ already has a class partition alignment, and the application of an entrywise non-linearity satisfying the hypothesis of the theorem, followed by a rotation, preserves it. To be more clear, $\sigma_{L}^{-1}(Y)$ satisfies DNC1, and the rotation preserves this property.
>
> **Additional comments**
> 1. The reviewer is correct; we updated equations 1 and 2 without biases, consistently with the rest of the paper.
> 2. We fixed this typo.
>
> We continue with the rest of the answers in the next response.

---

> > ### Author Response · Authors · 2026-05-08
> > **Official Comment by Authors 2/2**
> >
> > 3. We thank for noticing the typo regarding $z^{l-1}$, it has been corrected in the revised version. We opted for using the matrix notation in the proof of Proposition 1 to avoid any potential confusion. Note that the derivative of a vector valued function with respect to matrix parameters, corresponds to a three mode tensor (i.e., $\frac{\partial f(\Theta,x)}{\partial W_l}$ is the Jacobian of $f$ with respect to $W$, which is a linear map $\mathbb R^{n_{l+1} \times n_l} \to \mathbb R^{n_L}$, therefore the transposed (adjoint) is a map $\frac{\partial f(\Theta,x)}{\partial W_l}^\top: \mathbb R^{n_L} \to \mathbb R^{n_{l+1} \times n_l}$). As a second note, the order of the chain rule is correct, as what we represent is the gradient $\nabla_{W_l} \ell(f(\Theta,x),y)$ (which is an element of the vector space $\mathbb R^{n_{l+1} \times n_l}$), and not the differential $\partial_{W_l}\ell(f(\Theta,x),y)$ (which is an element of the dual space of $\mathbb R^{n_{l+1} \times n_l}$, i.e., a linear map $\mathbb R^{n_{l+1} \times n_l} \to \mathbb R$). In particular, in an inner product space, the gradient is defined as the unique vector $\nabla_{W_l} \ell(f(\Theta,x),y)$ such that:
> > $$\langle \nabla_{W_l} \ell(f(\Theta,x),y), u \rangle = \partial_{W_l}\ell(f(\Theta,x),y)[u],\quad \forall u \in \mathbb R^{n_{l+1} \times n_l}.$$
> > For this reason, if we have the composition of two functions $\tilde \ell = \ell \circ f$ ($\ell: \mathbb R^d \to \mathbb R$ scalar and $f: \mathbb R^{n \times m} \to \mathbb R^{d}$ vector valued), we have
> > $$
> > \langle \nabla \tilde \ell(W), u \rangle = d\tilde \ell(W)[u] = d\ell(f(W)) df(W)[u] = \langle \nabla \ell(f(W)),df(W)[u] \rangle = \langle df(W)^\top \nabla \ell(f(W)),u \rangle
> > $$
> > which implies that the chain rule for gradients is indeed "reversed", i.e., $\nabla \tilde \ell(W) = df(W)^\top \nabla \ell(f(W))$. We hope this clarifies this point.
> > 4. Yes, we fixed the typo in Eq.12, and in the text, it is $l=2$. We also included a comment (in the last paragraph before subsection 4.1) to better clarify the rationale behind the theoretical argument and why we study a modified functional, i.e., the minimizers of the modified functional can be characterized exactly in closed form, while we use a perturbative argument in Theorem 4.6 to guarantee their stability when the disregarded terms are included.
> > 5. Equation 10 and the introductory section were meant to give the main ideas behind the development of the following theory in section 4, and from where the representability constraint in Eq.11 comes from. The meaning of equation 10 is in fact that, for each compatible couple $(Z^{l+1}, Z^l)$ satisfying Eq.11, the minimal norm weight mapping one to the other is unique and can be represented by the right-hand side of Eq.10. This allows us to study the minimal norm interpolator problem on feature space, by adding representability constraint (I). We added a sentence to make this point clearer in the revised manuscript (in the sentence before Eq.10 we now refer to minimal norm weights, and in the sentence after Eq.10 we remark that the couple must be representable).
> > 6. The problem of having the term $||\sigma_1^{-1}(Z^1)X^+||_F^2$ is not connected with the Moore-Penrose pseudoinverse (which exists for any matrix). The problem is that the layer-by-layer peeling on the first layer will encounter an additional constraint including the matrix $X$, which makes the minimizers not available in closed form. This is the reason why we study a modified functional (in that setting, the minimizers can be described exactly in the setting of Theorem 4.2), and then study their stability perturbatively as we did in Theorem 4.6.
> > 7. We fixed the typo; it is indeed $rank(A)/2C$.
> > 8. We thank the reviewer for pointing this out; we modified all instances to $\frac{L-1}{2}rank(Y)$.

---

### Decision · Action_Editor_3vkB · 2026-06-15

**Recommendation:** Accept as is

**Audience:**

Yes

**Audience Explanation:**

The relationship between representation learning, neural collapse, and the structure of learned weights is of broad interest to researchers working on deep learning theory and representation learning.

**Claims And Evidence:**

Yes

**Claims Explanation:**

The paper aims to establish a tight connection between neural collapse, particularly variability collapse, and the low-rank bias of weight matrices. The claim is supported by theoretical analysis, which constitutes the main contribution of the paper, as well as empirical results that are broadly consistent with the theory.